# TENSOR-VAR: VARIATIONAL DATA ASSIMILATION IN TENSOR PRODUCT FEATURE SPACE

## ABSTRACT

Variational data assimilation estimates the dynamical system states by minimizing a cost function that fits the numerical models with observational data. The widely used method, four-dimensional variational assimilation (4D-Var), has two primary challenges: (1) computationally demanding for complex nonlinear systems, and (2) relying on state-observation mappings, which are often impractical. Deep learning (DL) has been used as a more expressive class of efficient model approximators to address these challenges. However, integrating such models into 4D-Var remains challenging due to their inherent nonlinearities and the lack of theoretical guarantees for consistency in assimilation results. In this paper, we propose *Tensor-Var* to address these challenges using kernel Conditional Mean Embedding (CME). Tensor-Var characterizes system dynamics and state-observation mappings as linear operators in a feature space, where it enables a linear 4D-Var framework with a convex cost function. Furthermore, our method provides a new perspective to incorporate CME with 4D-Var, offering theoretical guarantees of consistent assimilation results between the original and feature spaces. To improve scalability, we propose a method to learn adaptive deep features (DFs) using neural networks within the Tensor-Var framework. Experiments on chaotic systems and global weather prediction with real-time observations demonstrate that Tensor-Var outperforms conventional and DL hybrid 4D-Var baselines in terms of accuracy while achieving efficiency comparable to the static 3D-Var method (code available at `https://anonymous.4open.science/r/Tensor-Var-F1E9/README.md`).

## 1 INTRODUCTION

Forecasting of dynamical systems is an initial value problem of practical significance. Many real-world systems, such as the ocean and atmosphere, are *chaotic*, which means minor errors of current estimations in computational models can lead to rapid divergence and substantial forecasting errors. In this regard, data assimilation (DA) (Law et al., 2015; Asch et al., 2016) uses observation data to continuously calibrate models, improving forecast accuracy.

Various DA methods have been proposed to deal with different types of observation data and system dynamics. Among these methods, 4D variational (4D-Var) data assimilation has been considered cutting-edge and effectively used in real-world applications like numerical weather prediction (NWP) systems (Browne et al., 2019; Milan et al., 2020). The 4D-Var minimizes a quadratic cost function that finds the optimal match between system states and observations (Asch et al., 2016). While effective in NWP, there are two critical limitations for their applications: (1) Numerical models for complex, nonlinear systems are often inefficient for real-time assimilation and forecasting. (2) Observations are often noisy, incomplete representations of the states, even without a known state-observation mapping, posing challenges in utilizing the observation. Efforts have integrated DL models to learn an observation (or inverse) model (Frerix et al., 2021; Wang et al., 2022; Liang et al., 2023), addressing the imperfect knowledge of observation models in 4D-Var. While these approaches improve observation utilization, they remain constrained by the complexities of numerical models and learned observation mappings, whereas our approach simplifies this by finding their linear representations. To improve computation efficiency, the state-of-the-art DL models (Vaswani et al., 2017; Chen et al., 2018; Li et al., 2020; Kovachki et al., 2023; Bocquet et al., 2024) are capable of constructing highly nonlinear mappings to surrogate dynamical systems and achieve notable successes in NWP (Bi et al., 2022; Lam et al., 2022; Kurth et al., 2023; Chen et al., 2023; Conti,

2024; Vaughan et al., 2024). However, integrating such models into optimization-based tasks, like 4D-Var, remains challenging due to their inherent nonlinearities (Janner et al., 2021; Bocquet, 2023). Using auto-differentiation (AD) of DL models in 4D-Var can reduce computational costs and has shown success in simple examples (Geer, 2021; Dong et al., 2022; Cheng et al., 2024); however, the accuracy of AD-derived derivatives remains a concern, and its complexity grows with system dimensions (Baydin et al., 2018). Recently, Xiao et al. (2024) applied the AD of a pre-trained weather forecasting model into 4D-Var, forming a self-contained DA framework for Global NWP. However, this approach relies on the well-designed pre-trained model for the forward models and may not generalize easily to other domains. Latent data assimilation (Peyron et al., 2021; Fablet et al., 2021; Melinc & Zaplotnik, 2023; Cheng et al., 2023; Fablet et al., 2023) addresses these challenges by performing DA in a learned low-dimensional latent space. While efficient, these approaches lack theoretical guarantees for the consistency of 4D-Var solutions between the latent and original spaces.

In this paper, we introduce *Tensor-Var*, a framework for learning linear representation of the DA systems by using kernel conditional mean embedding (CME). Unlike existing DL-based methods face challenges with nonlinearity and non-convexity, we propose a new perspective from CME to linearize nonlinear dynamics, resulting in a convex cost function in the feature space. To best of our knowledge, our work is first attempt to integrate CME into 4D-Var for linear representation with convex cost function, greatly improving optimization efficiency and convergence. To address the challenges from incomplete observations, we derive an inverse observation operator that incorporates histories to infer the system state, thereby improving accuracy and robustness. Moreover, we provide a theoretical analysis that demonstrates the existence of a linear representation of the system under kernel features and the consistency of 4D-Var solution across original and latent spaces. A key challenge in extending CME to practical variational DA is scalability. To overcome this, our approach learns adaptive deep features (DFs) that map data into a fixed-dimensional feature space, reducing the computational complexities. Our experiments on two chaotic systems and two global NWP applications demonstrate that Tensor-Var outperforms conventional and ML-hybrid variational DA baselines, including operational and DL-hybrid methods, in accuracy and computational efficiency, showing the advantages of linearizing the DA systems through Tensor-Var.

## 2 BACKGROUND AND PROBLEM FORMULATION

**Notation.** Let $S$ and $O$ be random variables representing the state and observation, with their realizations in corresponding compact sets with $s \in \mathbb{R}^{n_s}$ and $o \in \mathbb{R}^{n_o}$. A sequence of states over time steps from 1 to $t$ is denoted by $s_{1:t} = (s_1, \ldots, s_t)$ and same for observations $o_{1:t}$. The $\|\cdot\|$ denotes the 2-norm.

### 2.1 4D VARIATIONAL DATA ASSIMILATION

Consider a dynamical system in discrete-time comprising dynamical model and observation model:

$$s_t = F(s_{t-1}) + \epsilon_t^s, \text{ and } o_t = G(s_t) + \epsilon_t^o, \tag{1}$$

in which $F$ is the dynamical model that advances the state $s_t$ to $s_{t+1}$, and $G$ is the observation model that maps the $s_t$ to the observation $o_t$. The noise components $\epsilon_t^s, \epsilon_t^o$ are assumed to follow the zero mean Gaussian distributions with covariance matrices $Q$ and $R$. The objective of (weakly constraint) 4D-variational DA is to minimize a cost function:

$$J(s_{0:T}) = \|s_0 - s_0^b\|_{B^{-1}}^2 + \sum_{t=0}^{T} \|o_t - G(s_t)\|_{R^{-1}}^2 + \sum_{t=1}^{T} \|s_t - F(s_{t-1})\|_{Q^{-1}}^2, \tag{2}$$

in which $s_0^b$ is a prior guess for the initial state $s_0$, with $B$ as the background covariance matrix representing the uncertainty, i.e., $s_0 \sim \mathcal{N}(s_0^b, B)$. The second and third terms account for errors in the observation and dynamical models in equation 1.

In systems with nonlinear dynamics and observations, the cost function is typically non-convex with costly model evaluations, raising challenges in minimization of equation 2. Thus, we seek a space with a linear structure, in which the 4D-Var optimization can be efficiently solved. Kernel methods provide a framework for linearization by projecting data into an infinite-dimensional feature space (Jacot et al., 2018; Bevanda et al., 2024).

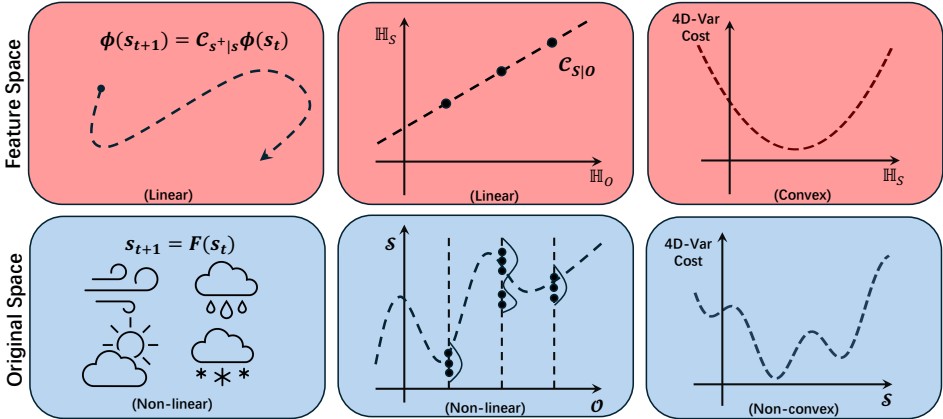

Figure 1: Demonstration of Tensor-Var: A DA system with nonlinear dynamical and observation models with non-convex cost function (bottom) can be represented linearly in feature space using kernel conditional mean embeddings that results in a convex cost function (Jacot et al., 2018) (top).

## 2.2 KERNEL CONDITIONAL MEAN EMBEDDING

A *Reproducing Kernel Hilbert Space* (RKHS) $\mathbb{H}_S$ with kernel $k_S$ is a Hilbert space, satisfying the reproducing property (Schölkopf & Smola, 2002). Let $k_S$ and $k_O$ be positive-definite (pd) kernels on RKHSs $\mathbb{H}_S$ and $\mathbb{H}_O$. We denote the kernel features as $\phi_O(o) = k_O(o, \cdot)$ and $\phi_S(s) = k_S(s, \cdot)$, referring the $\mathbb{H}_S$ and $\mathbb{H}_O$ as feature spaces. In this paper, we use the word kernel feature and the word feature interchangeably.

The kernel mean embedding of a distribution of $S$ is defined as the expectation of feature, $\mathbb{E}[\phi_S(S)] \in \mathbb{H}_S$ (Fukumizu et al., 2011). For characteristic kernels, these embeddings are injective, uniquely determining the probability distribution (Muandet et al., 2017). In addition to mean embedding, we will need the (uncentered) covariance operator (Baker, 1973) defined as $\mathcal{C}_{SO} = \mathbb{E}[\phi_S(S) \otimes \phi_O(O)]$, where $\otimes$ denotes the tensor product. This operator is also the embedding of the joint distribution as an element in the tensor product Hilbert space $\mathbb{H}_S \otimes \mathbb{H}_O$. The covariance operators extend the concepts of covariance matrices from finite dimensional spaces to infinite feature spaces.

**Conditional mean embedding.** To represent the dynamical and observation models in equation 1, the conditional mean embedding (CME) plays an important role. The CME of a conditional distribution of $S$ given $O = o$ is defined as the conditional expectation of kernel features $\mathbb{E}[\phi_S(S)|O = o]$. Under standard assumptions[1], there exists a linear operator $\mathcal{C}_{S|O} \colon \mathbb{H}_O \to \mathbb{H}_S$ such that $\mathbb{E}[\phi_S(S)|O = o] = \mathcal{C}_{S|O}\phi_O(o)$. Given independent and identically distributed samples $\{(s_i, o_i)\}_{i=1}^N$, an empirical estimate of the operator can be obtained as follows:

$$\hat{\mathcal{C}}_{S|O} = \hat{\mathcal{C}}_{SO}(\hat{\mathcal{C}}_{OO} + \lambda I)^{-1}, \tag{3}$$

where $\lambda$ is the regularization parameter, the $\hat{\mathcal{C}}_{SO}$ and $\hat{\mathcal{C}}_{OO}$ are empirical covariance operators, e.g. $\hat{\mathcal{C}}_{SO} = \frac{1}{N}\sum_{i=1}^N \phi_S(s_i) \otimes \phi_O(o_i)$, and $I$ denotes the identity matrix. Alternatively, the empirical estimate $\hat{\mathcal{C}}_{S|O}$ is equivalently to the minimizer of the following regression problem,

$$\hat{L}(\mathcal{C}) = \frac{1}{N}\sum_{i=1}^N \|\phi_S(s_i) - \mathcal{C}\phi_O(o_i)\|^2 + \lambda\|\mathcal{C}\|^2, \tag{4}$$

which offers a way to linearize the DA systems over the feature spaces.

## 3 METHODS

In this section, we introduce our *Tensor-Var* approach, which embeds 4D-Var into the kernel feature space and provides a theoretical analysis demonstrating the existence of linear dynamics with

---

[1](1) $\mathcal{C}_{SS}$ is injective, and (2) $\mathbb{E}[f(S)|O = o] \in \mathbb{H}_S$ for any $f \in \mathbb{H}_S$ and $o \in \mathbb{R}^{n_o}$.

consistent convergence between the original and feature space solutions. To effectively address incomplete observations, we propose an inverse observation operator that leverages consecutive historical observations. Finally, we also propose a method to learn adaptive deep features (DFs) using neural networks within the Tensor-Var framework, improving real-world applicability.

### 3.1 CME OF 4D-VAR IN RKHS

Having introduced the necessary tools for manipulating kernel embeddings, we now focus on learning the linearized models of the system in equation 1.

**CME of dynamical model.** Let $S^+$ be the one-step forward of $S$. Given the dynamical model $F$ in equation 1 and kernel feature $\phi_S$, the CME operator $\mathcal{C}_{S^+|S}$ can be recognized as the best linear approximation in the feature space $\mathbb{H}_S$ that minimize the regression residual $\mathbb{E}\big[\|\phi_S(S^+) - \mathcal{C}\phi_S(S)\|^2\big]$. Given finite data $\{(s_i^+, s_i)\}_{i=1}^N$ sliced from the system trajectory $s_{1:N+1}$, we can obtain the empirical estimate $\hat{\mathcal{C}}_{S^+|S}$ as equation 3 with theoretical supports of convergence (Fukumizu et al., 2013; Klus et al., 2020). The CME operator $\hat{\mathcal{C}}_{S^+|S}$ effectively characterizes the system dynamics as a linear model and simplifies the 4D-Var as a convex optimization in the feature space $\mathbb{H}_S$.

**CME of inverse observation model.** Analogous to the dynamical model, the observation model $G$ in equation 1 can be linearized by the CME operator $\mathcal{C}_{O|S}$, which has been used as observation models for filtering algorithms (Song et al., 2009; Fukumizu et al., 2013; Kanagawa et al., 2016; Gebhardt et al., 2019). Most approaches assume a complete observation setting, where the observations can fully determine the state. In practice, observations are often incomplete representation of state, with $n_s > n_o$, leading to underdetermined systems (Liu et al., 2022). In such systems, the lack of a bijective mapping between the state and observation spaces means that observations cannot uniquely determine the system's state. As a result, the optimization problem in 4D-Var may produce sub-optimal solutions (Asch et al., 2016). It is information-theoretically impossible to distinguish any two mixtures of states based on a single-step observation if $n_s > n_o$. By introducing the past $m$ consecutive observations as history $h_t = o_{t-m-1:t-1} \in \mathbb{R}^{m \times n_o}$, the joint information from history and current observation is enough to estimate the system state. The choice of history length is critical: too short lacks sufficient information, while too long is inefficient. Empirically, we performed an ablation study to assess the effects of history length, as detailed in the subsection 4.4.

To effectively incorporate history, we introduce another kernel feature $\phi_H$ with an induced RKHS $\mathbb{H}_H$. The space $\mathbb{H}_{OH} = \mathbb{H}_O \otimes \mathbb{H}_H$ is called a tensor product RKHS on $O \times H$ with associated kernel feature $\phi_{OH}(o, h) = \phi_O(o) \otimes \phi_H(h)$. As shown in (Song et al., 2013; Muandet et al., 2017), the CME operator can be extended to high-order features, allowing us to embed the joint distributions over $s_t$, $o_t$, and $h_t$ into the feature space (Song et al., 2009; 2013). The tensor product feature $\phi_{OH}$ captures the high-order dependencies between observations and history. The CME operator $\mathcal{C}_{S|OH}$ is the linear inverse observation model that minimizes the state estimation error given observations and history. Given dataset $\{(s_i, o_i, h_i)\}_{i=1}^N$, the empirical counterpart $\hat{\mathcal{C}}_{S|OH} = \hat{\mathcal{C}}_{SOH}(\hat{\mathcal{C}}_{(OH)(OH)} + \lambda I)^{-1}$ follows the same way as equation 3. The $\hat{\mathcal{C}}_{SOH}$ is the empirical high order tensor, e.g., $\hat{\mathcal{C}}_{SOH} = \sum_{i=1}^N \phi_S(s_i) \otimes \phi_O(o_i) \otimes \phi_H(h_i)$.

Our approach is closely related to delay embeddings, a well-established method in dynamical system theory for reconstructing attractors and state spaces from sequential data (Sauer et al., 1991; Krämer et al., 2021). As shown by Takens' embedding theorem (Takens, 2006), the dynamics of a system can be reconstructed in a higher-dimensional space using historical observations. Parallel to Takens' embedding theorem, recent theoretical advancements (Uehara et al., 2022) have focused on partial observability from a learning theory perspective. Liu et al. (2022) showed that the required history length for effective state reconstruction is determined by the complexity of the dynamical system.

**Feature space 4D-Var.** Using the kernel features, we linearize the original nonlinear dynamics and observations in the feature space $\mathbb{H}_S$. This transformation enables us to reformulate the 4D-Var optimization objective equation 2 into the feature space and optimize over a sequence of elements $z_{0:T}$ in feature space $\mathbb{H}_S$:

$$\min_{z_{0:T}} \|z_0 - \phi_S(s_0^b)\|_{\mathcal{B}^{-1}}^2 + \sum_{t=0}^{T} \|z_t - \mathcal{C}_{S|OH}\phi_{OH}(o_t, h_t)\|_{\mathcal{R}^{-1}}^2 + \sum_{t=0}^{T-1} \|z_{t+1} - \mathcal{C}_{S^+|S}z_t\|_{\mathcal{Q}^{-1}}^2, \quad (5)$$

where the $\mathcal{B}$, $\mathcal{R}$, and $\mathcal{Q}$ are the covariance operators for the background error, observation error, and model error in the feature space. In this work, we estimated the three operators as the empirical error covariance matrices from the training dataset (the explicit estimation can be found in Appendix B.1). As a result, our approach linearizes the original nonlinear dynamical and observation models, reducing the problem to solving a linear 4D-Var problem, with a quadratic cost function with linear dynamics $z_{t+1} = \mathcal{C}_{S+|S} z_t$. We present the pseudo-algorithms for CME of 4D-Var and performing Tensor-Var, detailed in Algorithms 2 and 3 in Appendix B.2.

### 3.2 LEARNING THE DEEP FEATURES WITHIN TENSOR-VAR

Using the pre-determined kernel features has theoretical guarantees. However, it maps data into an infinite-dimensional feature space, e.g., radial basis function kernel, making empirical estimation challenging due to polynomial scaling with sample size. On the other hand, these feature maps struggle with irregular or high-dimensional data, often resulting in poor performances. Learned deep features (DFs) have emerged as alternatives to generic kernel features (Xu et al., 2022; Kostic et al., 2023; Shimizu et al., 2024), project the data into a fixed-dimensional feature space, similar to methods like low-rank approximation (Williams & Seeger, 2000). To improve the scalability, we integrate DFs with the Tensor-Var framework and validate their effectiveness through experiments.

**Learning the state feature.** Recall that the CME operator $\mathcal{C}_{S+|S}$ is the best linear approximation of the system dynamics in the feature space. We propose to jointly learn the feature $\phi_{\theta_S} : \mathbb{R}^{n_s} \to \mathbb{R}^{d_s}$ with $\hat{\mathcal{C}}_{S+|S}$ by minimizing the loss $L(\theta_S) = \min_{\mathcal{C} \in \mathbb{R}^{d_s \times d_s}} L(\mathcal{C}, \theta_S) = \mathbb{E}\left[\|\phi_{\theta_S}(s^+) - \hat{\mathcal{C}}_{S+|S}\phi_{\theta_S}(s)\|^2\right]$. Predictions using $\hat{\mathcal{C}}_{S+|S}$ are in the feature space; however, for DA problems, reconstruction to the original state space is required, which is known as the preimage problem (Honeine & Richard, 2011). Here, we learn an inverse feature $\phi_{\theta_S'}^\dagger$ to solve the preimage problem during training, avoiding repeated optimization whenever computing preimages. The final training loss is the combination of the two terms as

$$L(\theta_S, \theta_S') = \mathbb{E}\left[\|\phi_{\theta_S}(s^+) - \hat{\mathcal{C}}_{S+|S}\phi_{\theta_S}(s)\|^2\right] + w\,\mathbb{E}\left[\|s - \phi_{\theta_S'}^\dagger\big(\phi_{\theta_S}(s)\big)\|^2\right], \tag{6}$$

where $w \in [0, 1]$ is a weighting coefficient, and $\hat{\mathcal{C}}_{S+|S}$ is computed as the CME over training batches. Note that using the DFs corresponds to a linear kernel in the learned feature space, where $k(s_i, s_j) = \phi_{\theta_S}(s_i)^T \phi_{\theta_S}(s_j)$.

**Learning the observation and history features.** Similar to learning the state feature, $\mathcal{C}_{S|OH}$ is the minimizer of regression problem mapping the tensor product of observation and history features to the state feature. In this phase, we learn the DFs for observation $\phi_{\theta_O} : \mathbb{R}^{n_o} \to \mathbb{R}^{d_o}$, history $\phi_{\theta_H} : \mathbb{R}^{n_h} \to \mathbb{R}^{d_h}$, and $\mathcal{C}_{S|OH}$ jointly with the loss function:

$$L(\theta_O, \theta_H) = \mathbb{E}\left[\|\phi_{\theta_S}(s) - \hat{\mathcal{C}}_{S|OH}[\phi_{\theta_O}(o) \otimes \phi_{\theta_H}(h)]\|^2\right],$$

where $\hat{\mathcal{C}}_{S|OH}$ is computed similarly over training batches in parallel, with the $\otimes$ denotes Kronecker product in practice. The learning procedure of DFs is summarized in Algorithm 1.

### 3.3 THEORETICAL ANALYSIS.

In Section 3.1, we discuss how the dynamical system $F$ in equation 1 can be embedded as a linear system in the feature space. However, two important questions remain: *1) Does such linear dynamical system exist? 2) Are the solutions of original and feature space 4D-Var consistent?* In this section, we provide affirmative answers to both questions using the theory of Kazantzis-Kravaris/Luenberger (KKL) observers (Andrieu & Praly, 2006). We give a road-map of the theoretical analysis with main result and refer readers to Appendix A for the details.

Under the mild assumptions that (1) the dynamical model $F$ is first-order differentiable and (2) the kernel features are all first-order differentiable, the kernel feature $\phi_S$ satisfies the necessary conditions as the state transformation in the KKL observer framework. This transformation enables us to represent the nonlinear dynamical system as a linear system in a higher-dimensional feature space, as established by KKL observer theory (Tran & Bernard, 2023). This result confirms the

Table 1: Comparison of DA performances. All baseline methods use the strong-constraint 4D-Var objective, while our approach uses the weak-constraint 4D-Var objective. Evaluation times are reported as each assimilation window's mean and standard deviation. Our method consistently outperforms the baselines across all benchmark domains.

| Domain | Algorithm | NRMSE (%) | Evaluation time ($10^{-2}$s) |
|---|---|---|---|
| Lorenz 96 $n_s = 40, n_o = 8$ | 3D-Var | $14.17 \pm 0.93$ | $12.59 \pm \mathbf{0.39}$ |
| | 4D-Var | $12.27 \pm 1.41$ | $210.52 \pm 3.87$ |
| | Frerix et al. (2021) | $9.89 \pm 1.63$ | $167.43 \pm 1.33$ |
| | Ours | $\mathbf{8.32 \pm 0.87}$ | $\mathbf{12.51} \pm 1.97$ |
| Lorenz 96 $n_s = 80, n_o = 16$ | 3D-Var | $15.19 \pm 1.09$ | $\mathbf{19.38 \pm 0.37}$ |
| | 4D-Var | $12.38 \pm 1.11$ | $322.21 \pm 5.73$ |
| | Frerix et al. (2021) | $10.79 \pm \mathbf{0.57}$ | $286.11 \pm 2.43$ |
| | Ours | $\mathbf{9.04} \pm 1.32$ | $21.09 \pm 0.79$ |
| Kuramoto-Sivashinsky $n_s = 128, n_o = 32$ | 3D-Var | $17.64 \pm 1.27$ | $\mathbf{16.48 \pm 1.17}$ |
| | 4D-Var | $15.46 \pm 1.07$ | $94.83 \pm 3.89$ |
| | Frerix et al. (2021) | $10.25 \pm 1.34$ | $63.28 \pm 1.91$ |
| | Ours | $\mathbf{9.69} \pm 1.56$ | $19.58 \pm 1.23$ |
| Kuramoto-Sivashinsky $n_s = 256, n_o = 64$ | 3D-Var | $16.66 \pm 0.69$ | $\mathbf{15.81 \pm 0.92}$ |
| | 4D-Var | $10.67 \pm 0.62$ | $95.68 \pm 1.35$ |
| | Frerix et al. (2021) | $8.87 \pm 0.55$ | $68.39 \pm 1.23$ |
| | Ours | $\mathbf{4.31 \pm 0.19}$ | $17.37 \pm 1.36$ |

existence of such a linear system, thus answering question 1). The KKL observer theory provides a theoretical foundation for our approach, bridging nonlinear dynamics and linear 4D-Var methods (Andrieu & Praly, 2006). A detailed derivation proving that $\phi_S$ satisfies the conditions as the state transformation of the KKL observer can be found in Appendix A.

We consider the nonlinear system in equation 1 within a compact state space and assume that the cost function in 2 has a unique solution. Given that $\phi_S$ is a state transformation in the KKL observer, the system in the original state space can be represented linearly in the feature space. The solution in the feature space have consistent convergence to the unique solution of original 4D-Var problem, minimizing with respect to the cost function 5, answering the question 2). A formal theorem with detailed proofs can be found in Theorem A.4 in Appendix A.2.

## 4 NUMERICAL EXPERIMENTS

To evaluate our proposed method, the comparison is conducted on a series of benchmark domains, representing optimization problem equation 2 of increasing complexity, including (1) the Lorenz 96 system (Lorenz, 1996) with 40 and 80 state dimensions. (2) The Kuramoto-Sivashinsky (KS) equation: a fourth order nonlinear PDE system (Papageorgiou & Smyrlis, 1991) with 128 and 256 dimensions, representing different spatial resolution. For both systems, we use a nonlinear observation model $o = G(s) = 5\arctan(s\pi/10) + \epsilon$, where $\epsilon$ is white noise with a standard deviation of 0.01 times the standard deviation of state variable distribution and only $20\%$ states can be observed. To assess the practical applicability of Tensor-Var, we evaluate its performance in global medium-range weather forecasting (i.e., 3-5 days) by using a subset of the ECMWF Reanalysis v5 (ERA5) dataset for training and testing, with further details in subsection 4.2 and (Rasp et al., 2024). Furthermore, we incorporate observations from real-time weather satellites into the NWP experiment with higher spatial-resolution, showcasing the practical utility of Tensor-Var in subsection 4.3.

**Baselines.** We compare our method against several baseline approaches: (1) 3D-Var with a known observation model, (2) a model-based 4D-Var algorithm that assumes known dynamical and observation models, (3) a learned inverse observation model with the known dynamical model, as proposed by Frerix et al. (2021), and (4) In the two NWP problems, we include two more baselines: Latent 3D-Var and Latent 4D-Var (Cheng et al., 2024), which perform variational DA in a compressed latent space learned via autoencoder, with Latent 4D-Var additionally learning the dynamics in the latent space. These competitive baselines cover both operational Var-DA and ML-hybrid Var-DA methods.

## 4.1 EVALUATION AND RESULTS.

In each experiment, we measure the quality of assimilation using the Normalized Root Mean Square Error (NRMSE) $\sqrt{\frac{1}{T} \sum_{t=0}^{T} \|\hat{s}_t - s_t\|^2}/(s_{\max} - s_{\min})$ over the assimilation window, where $s_{\max}, s_{\min}$ are the maximum and minimum state value in training dataset. These results and the average evaluation time on the AMD 7980X CPU for each algorithm are reported in Table 1. All metrics are evaluated 20 times with different initial conditions, reporting the mean and standard deviation. For Tensor-Var, the history length $m$ is selected using a cross-validation approach as an ablation study in subsection 4.4, and the objective function in feature space is minimized using quadratic programming methods, implemented via CVXPY (Diamond & Boyd, 2016). Baselines use the L-BFGS method (Nocedal & Wright, 2006) with 10 history vectors for the Hessian approximation. For 4D-Var baselines, the adjoint method is not used, and the strongly constrained 4D-Var is applied. The background state $s_b$ is set to the average state of the training set.

As shown in Table 1, our method consistently outperforms the other baseline methods in all metrics. In all four tasks, Tensor-Var achieves the lowest mean and standard deviation of NRMSE for assimilation accuracy, demonstrating strong generalization from the Lorenz-96 systems to the more complex KS systems. The better performances are attributed to the linearization of the dynamics and the history-augmented inverse observation operator, which makes the 4D-Var optimization convex with more reliable convergence to the global optimum in the feature space. The larger NRMSE errors observed in the model-based 3D-Var and 4D-Var methods are due to that incomplete observations lead to the uncontrolled errors in the unobserved state dimensions. Although the ML-hybrid 4D-Var method (Frerix et al., 2021) employs a learned inverse observation model, it struggles with generalization due to the incomplete observations, leading to poor performance on test data. By incorporating historical information, our approach more effectively controls estimation errors in the unobserved state dimensions. This improvement is clearly demonstrated in Figures 8 and 9 in Appendix C.1.2 and C.1.2, which qualitatively compares assimilation performance in the Lorenz-96 and KS systems. In terms of computational efficiency, the linearity of Tensor-Var makes it faster than all 4D-Var baselines while also outperforming or being comparable to the static 3D-Var method, attributing to the reduced complexity of solving convex optimization problems in the feature space.

## 4.2 GLOBAL NWP

Next, we consider a global NWP problem. The European Centre for Medium-Range Weather Forecasts (ECMWF) Atmospheric Reanalysis (ERA5) dataset (Hersbach et al., 2020) provides the best estimate of the dynamics of the atmosphere covering the period from 1940 - present. For a proof of concept, the 500hPa geopotential, 850 hPa temperature, 700 hPa humidity, and 850 hPa wind speed (meridonal and zonal directions) at $64 \times 32$ resolution is considered here. The data is sourced from the WeatherBench2 repository (Rasp et al., 2024). Observations are sampled randomly from the grid with a 15% spatial coverage and with additive noise (0.01 times the standard deviation of the state variable) (see Figure 2). Latent dimensions of the two baseline methods are set to match the feature dimension $d_s$ in our approach. The evaluation metric is the area-weighting RMSE over grid points (see more details in Appendix C.3). We trained all the models on ERA5 data from 1979-01-01 to 2016-01-01 and tested on data post-2018, with a qualitative evaluation shown for 2018-01-01 00:00 in Figure 2.

Figure 3 (left 1-5) shows the distributions of NRMSE (%) across different atmospheric variables (z500, t850, q700, u850, v850) for Latent 3D-Var, Latent 4D-Var, and Tensor-Var, evaluated on a test dataset consisting two years of data from 2018-01-01 00:00 to 2020-01-01 00:00. Our approach consistently achieves the lowest mean and standard deviation of NRMSE for all variables, demonstrating improved performance in both assimilation accuracy and robustness. The latent space of 3D-Var and 4D-Var is learned by autoencoder purely based on the reconstruction loss without considering system dynamics. This weakens the forecasting abilities of the dynamical systems, introducing extra errors in the 4D-Var optimization compared to the 3D-Var. In contrast, our approach jointly learns the feature space representation and linear system dynamics, resulting in more accurate forecasting and improved 4D-Var performances.

The rightmost bar plot in Figure 3 shows evaluation times on an Nvidia RTX-4090 GPU. Tensor-Var outperforms Latent 3D-Var and Latent 4D-Var, demonstrating its better computational efficiency over

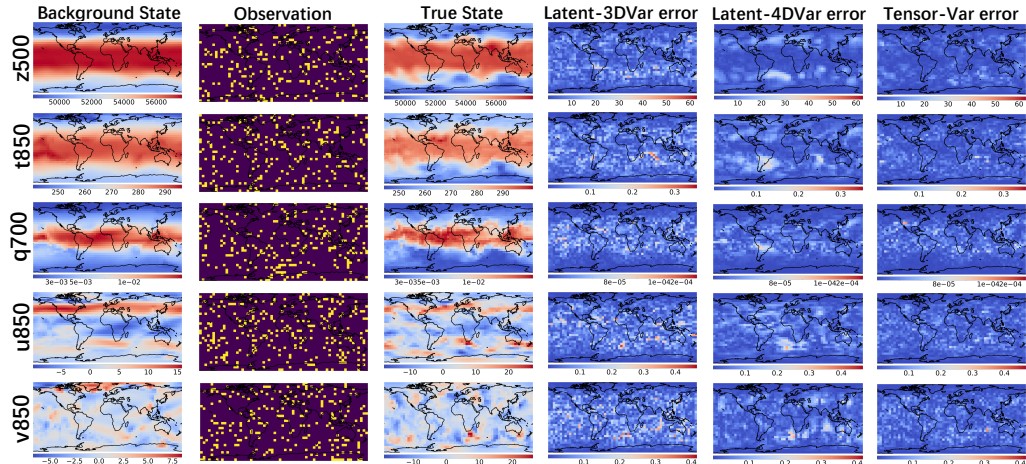

Figure 2: Visualization of assimilation results for five variables from ERA5 data at time 2018-01-01 00:00. Each column (from left to right) displays the background state, observations, true state, and errors for Latent-3DVar, Latent-4DVar, and Tensor-Var. The reported error was a weighted absolute error in each pixel; see Appendix C.3 for more details.

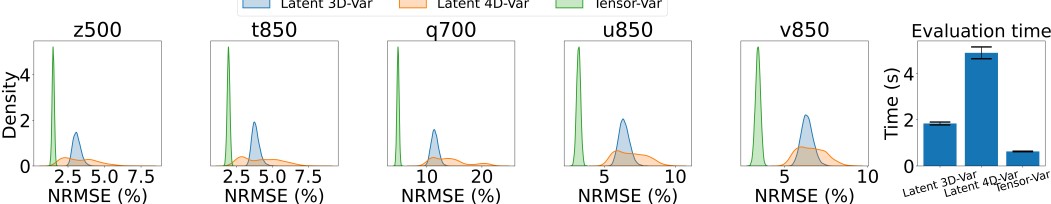

Figure 3: Comparison of distribution of NRMSE (%) across different atmospheric variables (z500, t850, q700, u850, v850) for Latent 3D-Var, Latent 4D-Var, and Tensor-Var. The rightmost bar plot shows evaluation times and error bars indicate the standard deviation for evaluation time.

the data-driven Var-DA approaches. This is because our deep features are used only for mapping data into the feature space rather than being directly involved in the gradient-based optimization via AD. In addition to the assimilation results, we evaluate the forecasting quality of Tensor-Var based on the assimilated state, quantatitive results can be found in Appendix C.3 Figure 10.

### 4.3 ASSIMILATION FROM SATELLITE OBSERVATIONS

In the final experiment, in contrast to the randomly selected observation locations in section 4.2, we now consider a more realistic DA problem in global NWP by incorporating the location of satellite tracks. Weather satellites are a critical source of observational data for DA in global NWP; whilst introducing this observation is nontrivial as it incorporates further data fusion of the dynamic observation locations of the satellites and the spatial-temporal sparsity of satellite tracks. The spatial resolution of the underlying grid is even increased to $240 \times 121$ in this case.

An example of satellite tracks and observation distribution is shown in Figure 4. We extract satellite track data (latitude and longitude coordinates) from CelesTrak[2] for the same periods as Section 4.2, matching it with ERA5 data to generate practical observations. These observations include satellite locations within two hours before the assimilation time, sampled at half-hour intervals, with an average coverage of approximately 6%, see Figure 6. Other experimental settings, such as data volume, training/testing periods, variables, and the 4D-Var window, align with Section 4.2.

---

[2]CelesTrak provides publicly accessible orbital data for a wide range of satellites, including those with meteorological sensors at `www.celestrak.com`. The data include positional details, and temporal information, allowing for accurate real-time satellite tracking.

In this experiment, we evaluate our proposed method in an online operational scenario, where the DA is continuously applied. To assess the uncertainties, all methods are evaluated 10 times by randomly sampling sequences from the test dataset. We present the mean and standard deviation of NRMSEs in Figure 5 over a time-horizon of 7-days. Tensor-Var consistently achieves the lowest mean and standard deviation, demonstrating its robustness in a large-scale system with practical observations. Figure 6 presents the qualitative assimilation results for the variable z500 with observations; results for other variables are provided in Appendix C.4. Dynamic satellite observations impact assimilation accuracy, with clustered observations near the equator from geosynchronous satellites substantially reducing errors in the corresponding regions. The results demonstrate

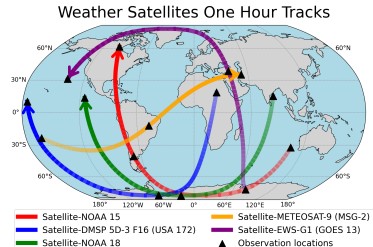

Figure 4: Example of selected satellite tracks over a one-hour horizon, with observations (black triangles) sampled at half-hour intervals.

that Tensor-Var excels in accuracy and robustness, when handling large-scale systems with practical observations, showing its potential for applications in operational DA and forecasting.

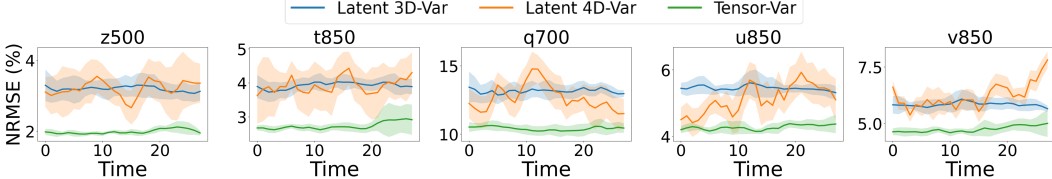

Figure 5: Comparison of assimilation NRMSE (%) over a 7-day horizon for five atmospheric variables from Latent 3D-Var, Latent 4D-Var, and Tensor-Var. Each time-step represents a 6-hour interval.

## 4.4 ABLATION STUDY

To support our empirical results, we conducted ablation studies on the 40- and 80-dimensional Lorenz-96 system to investigate (1) the effect of history length in learning the operator $\hat{\mathcal{C}}_{S|OH}$, and to explore (2) the effect of DFs $\phi_S$ with different feature dimensions.

**Effect of history length.** We explored the effect of history length $m$ on learning the inverse observation operator and its impact on state estimation accuracy. According to the theory in (Liu et al., 2022), the history length can be chosen as $m \propto \log(n_s)^3$. The ablation study was conducted by scaling $m$ proportionally to $m \approx C \log n_s$ where the constant $C$ was adjusted. The feature dimensions $d_s, d_o, d_h$ are fixed to be the same as the experiments in subsection 4.1.

Table 3 shows that incorporating history ($C > 0$) significantly improves state estimation accuracy, with NRMSE decreasing as $C$ increases. However, the improvements become marginal when increasing $C$ beyond a certain point (around $C = 4$), suggesting diminishing improvements for larger history lengths. This indicates a trade-off, where increasing the history length beyond a certain threshold yields little additional benefit in state estimation.

Table 2: Comparison of different history lengths, with NRMSE as the metric for state estimation accuracy. $C = 0$ indicates that no history is incorporated.

|  | $C = 0$ | $C = 1$ | $C = 2$ | $C = 4$ | $C = 8$ |
|---|---|---|---|---|---|
| $n_s = 40$ | $11.7 \pm 3.4$ | $9.3 \pm 2.8$ | $7.7 \pm 1.4$ | $7.7 \pm 0.9$ | $7.5 \pm 0.5$ |
| $n_s = 80$ | $10.8 \pm 2.6$ | $8.2 \pm 1.7$ | $7.4 \pm 0.9$ | $7.1 \pm 0.9$ | $7.0 \pm 0.8$ |

**Effect of DFs.** In this experiment, we compared the DFs with different dimensions $d_s = 20, 40, 60, 80$ and Gaussian kernel feature. To scale up the Gaussian kernel, we applied Nyström approximation and kernel PCA (See Appendix C.5 for details). All three features are Gaussian kernels. The pre-image of the system state was learned using kernel ridge regression on the low-dimensional representations. As shown in Table 3, Gaussian kernel performance was close to the best DFs in

---

[3]Please note that the result omits the class of systems with exponential dependency on the history length.

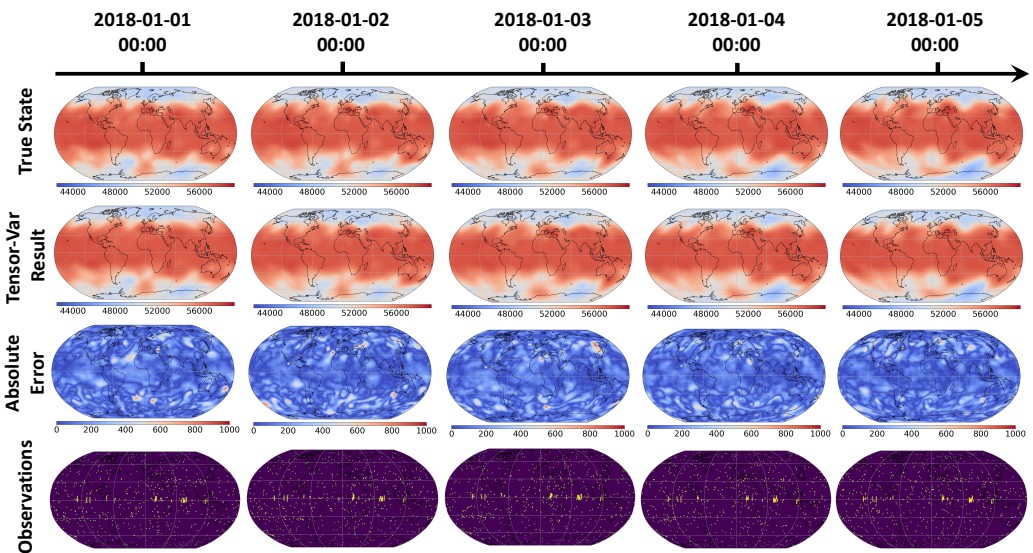

Figure 6: Visualization of continuous assimilation results, absolute errors, and observation locations for z500 (geopotential), starting from 2018-01-01 00:00. The observation coverage, defined as the ratio of the number of observations to the number of grid points, is 6.37%.

$n_s = 40$ but degraded in $n_s = 80$. In lower-dimensional problems, pre-determined kernel features were more robust than DFs, but their performance can degrade with increasing system dimensionality and complexity. DFs with $d_s = 60, 120$ consistently performed well, while $d_s = 20, 40$ performed poorly, reflecting the trade-offs among data size, parameter count, and system dimension.

Table 3: Comparison of different features. The metric is NRMSE (%) over the assimilation window

|  | $d_s = 20$ | $d_s = 40$ | $d_s = 60$ | $d_s = 120$ | Gaussian kernel |
|---|---|---|---|---|---|
| $n_s = 40$ | $16.7 \pm 2.1$ | $14.1 \pm 1.3$ | $\mathbf{8.3} \pm 0.9$ | $9.7 \pm 0.7$ | $8.5 \pm \mathbf{0.5}$ |
| $n_s = 80$ | $17.3 \pm 2.6$ | $16.8 \pm 2.2$ | $11.4 \pm 0.9$ | $\mathbf{9.0} \pm \mathbf{0.9}$ | $14.3 \pm 1.4$ |

## 5 CONCLUSION

In this paper, we introduced the Tensor-Var method, using kernel conditional mean embedding to transform nonlinear dynamical and observation models in 4D-Var into a linear framework, making the optimization both tractable and computationally efficient. By learning adaptive deep features, Tensor-Var addresses the scalability typically associated with traditional kernel methods. Our inverse observation operator, which incorporates historical observations, improving the accuracy and robustness with incomplete observations. Experiments on two chaotic systems and global weather forecasting show that Tensor-Var outperforms state-of-the-art hybrid ML-DA models in both accuracy and efficiency.

**Limitations.** Our method requires access to exact or re-analyzed system states to learn the dynamical and observation models, which may not be feasible in practical applications. A future direction would be to learn these models directly from observations and calibrate the learned dynamics in the feature space using Tensor-Var. In addition, our simplified error covariance matrices in 4D-Var may not fully capture system correlations and uncertainties. Future work could focus on improving the design of error covariance matrices within the Tensor-Var framework to enhance assimilation and performance.

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

## A  THEORETICAL ANALYSIS

In this section, we provide a theoretical convergence analysis of Tensor-Var, drawing on concepts from control theory and contraction analysis. We begin by introducing comparison functions, including class $\mathcal{K}$ and $\mathcal{KL}$ functions, as well as Lyapunov functions. Using these tools, we examine the convergence of Tensor-Var through a differential equation, demonstrating monotonic contraction based on Lyapunov direct method. Furthermore, by using comparison functions, we show that contraction in the feature space implies contraction in the original space.

**Assumption 1.** To conduct a formal convergence analysis of Tensor-Var, we make a mild assumption regarding the first-order differentiability of the dynamical system $F$ in equation 1 with time derivative $\dot{s} = f(s)$, a standard assumption in convergence studies (Sastry, 2013).

**Assumption 2.** We require that the kernel possess a well-defined first-order derivative, as the convergence analysis is performed in the feature function space. This assumption is common in kernel methods and is satisfied by many widely-used kernels, such as the Gaussian, Fourier, Matérn, and Laplace kernels (Berg et al., 1984; Schölkopf & Smola, 2002).

### A.1  NOTATIONS AND TECHNICAL LEMMAS

**Definition A.1 (Class $\mathcal{K}$ function (Gajic & Qureshi, 2008).)**
*A continuous function $\alpha : [0, a] \to [0, \infty)$ is said to belong to class $\mathcal{K}$ if it is strictly increasing and $\alpha(0) = 0$. It is of class $\mathcal{K}_\infty$ if $\alpha(\infty) = \infty$ and $\alpha(r) \to \infty$ as $r \to \infty$.*

**Definition A.2 (Class $\mathcal{KL}$ function (Gajic & Qureshi, 2008).)**
*A continuous function $\beta : [0, a] \times [0, \infty) \to [0, \infty)$ is said to belong to class $\mathcal{KL}$ if each fixed $t$, the mapping $\beta(r, t)$ belonging to the class $\mathcal{K}$ with respect to $r$ and, for each fixed $a$ the mapping $\beta(a, t)$ is decreasing with respect to $t$ and, $\beta(a, t) \to 0$ as $t \to \infty$.*

The $\mathcal{K}$ and $\mathcal{KL}$ are two classes of comparison functions, we can use the comparison function to analyze the monotone contraction in both spatial and temporal horizons.

**Definition A.3 (Lyapunov Stability (Gajic & Qureshi, 2008))**
*If the Lyapunov function $V$ is globally positive definite, radially unbounded, the equilibrium isolated and the time derivative of the Lyapunov function is globally negative definite:*

$$\frac{dV}{dt}(x) < 0, \qquad \forall \mathbb{R}^n \setminus \{0\}, \tag{7}$$

*then the equilibrium is proven to be globally asymptotically stable. The Lyapunov function is a class $\mathcal{K}$ function, which satisfying the condition as follow*

$$\alpha_1(\|x\|) \leq V(x) \leq \alpha_2(\|x\|), \quad , \forall x \in [0, \infty). \tag{8}$$

**Lemma A.1 (Hurwitz stability criterion (Parks, 1962))**
*A square matrix $\mathcal{A} \in \mathbb{R}^{n \times n}$ is said to be Hurwitz stable if all the eigenvalues of $\mathcal{A}$ have strictly negative real parts, i.e., for every eigenvalue $\lambda$ of $\mathcal{A}$,*

$$Re(\lambda) < 0. \tag{9}$$

*In other words, the real part of each eigenvalue of the matrix must lie in the left half of the complex plane.*

**Lemma A.2 (Hurwitz stability criterion via Lyapunov function (Sastry, 2013))**
*Give a candidate Lyapunov function for linear dynamics as*

$$V(x) = x^T P x, \quad \frac{dx}{dt} = \mathcal{A}x \tag{10}$$

*where $P$ is symmetric, positive definite matrix; $\mathcal{A}$ governs the evolution of dynamics. For the system to be stable, the time derivative $\frac{dV}{dt}$ must be negative definite, i.e., $\frac{dV}{dt} < 0$ for all $x \neq 0$. This means that:*

$$\frac{dV}{dt}(x) = \frac{d}{dt}(x^T P x) = x^T(\mathcal{A}^T P + P\mathcal{A})x < 0 \qquad (11)$$

*with*

$$\mathcal{A}^T P + P\mathcal{A} < 0, \qquad (12)$$

*where $(\mathcal{A}^T P + P\mathcal{A})$ is negative definite.*

Based on the convergence analysis and Lyapunov theory, a more generalized concept – *contraction metric* is needed to support our paper.

**Definition A.4 (Contraction Manchester & Slotine (2017))**
*Given the system $\frac{dx}{dt} = f(x,t)$, if there exists a uniformly bounded metric $M(x,t)$ (positive definite) such that*

$$\frac{dM}{dt} + \frac{\partial f}{\partial x}(x,t)^T M + M \frac{\partial f}{\partial x}(x,t) < -cM, \quad c > 0, \qquad (13)$$

*then we call the system contracting, and $M(x,t)$ is a contraction metric.*

### A.2    PROOF OF KEY THEOREMS

To adopt the convergence analysis of DA problem, we give a mild assumption on the smoothness of dynamics, the first-order derivative exists in the system 1.

**Theorem A.3 (Embedding and Consistent convergence)**
*Here, we consider system $s_{t+\Delta t} = F_{\Delta t}(s_t) = s_t + \int_t^{t+\Delta t} f(s_\tau)d\tau$ in equation 1 is first order differentiable with derivative $\frac{ds_t}{dt} = f(s_t)$ for $s \in \mathbb{R}^{n_s}$ with a unique equilibrium point as $s_*$. If there exists a embedding as $\phi_S(s) := [\phi_S^1(s_t), \dots, \phi_S^{d_s}(s_t)]^T$ with that $d_s \in \mathbb{N} \cup \infty$ satisfying the following properties:*

- ***a.** (embedding) For a finite $d_s$, the $\frac{\partial \phi}{\partial s}(s_t)$ is full-column rank; when $d_s$ is infinite, it is assumed to be rank-$d_s$ countably infinite, i.e. $\{\nabla\phi_S(s_t)\}$ is full-column rank with $\nabla\phi_S(s_t) = [\frac{\partial \phi_S^1}{\partial s}(s_t), \dots, \frac{\partial \phi_S^n}{\partial s}(s_t), \dots]^T$.*

- ***b.** (convergence) There exists Hurwitz matrix $\mathcal{A}$ verifying*

$$\frac{d\phi_S}{dt}(s_t) = \mathcal{A}\phi_S(s_t). \qquad (14)$$

  *Then, the equilibrium $s_*$ and $\phi_S(s_*)$ are global asymptotic convergence.*

*Proof.*

(Embedding.) The embedding property follows from RKHS theory. Since we restrict our analysis to a separable Hilbert space, it has a countable basis either finite or infinite. Thus, the dimension of the RKHS can be infinite, but the rank of the embedding is determined by a countable set of basis functions (Schölkopf & Smola, 2002).

(convergence.) According to the differential equation in Hilbert space, we have

$$\begin{aligned}
&\frac{d\phi_S}{dt}(s_t) \\
&= \frac{\partial \phi_S}{\partial s_t}(s_t) \cdot \frac{ds_t}{dt} \\
&= \nabla\phi_S(s_t)f(s_t) \\
&= \mathcal{A}\phi_S(s_t).
\end{aligned} \qquad (15)$$

The second line of equation 15 follows from chain rule, the final line represents the time derivative in Hilbert space, where $\mathcal{A}$ is a linear operator that governs the dynamics. Following the work from Romanoff (1947); Bobrowski (2016), we define $\mathcal{A}$ as:

$$\mathcal{A} := \lim_{t \to 0^+} \frac{\mathcal{C}_{S^+|S} - Id}{t}, \qquad (16)$$

where $\mathcal{C}_{S^+|S}$ is the conditional covariance operator between future and current states in the RKHS. Given the smoothness of the kernel function and the differentiability of the system dynamics, the linear operator $\mathcal{A}$ exists and well-defined in this context.

Since $s_*$ is the equilibrium point, we can derive a natural result that $f(s_*) = 0$. Invoking equation 15 (third line) we have $\frac{d\phi_S}{dt}(s_*) = 0$. Thus $\phi_S(s_*)$ is also a local equilibrium point in RKHS. From the embedding property of RKHS, we have a local injective map $\phi : \mathbb{R}^{n_s} \to \mathbb{R}^{d_s}$, which ensures that the convergence properties of the system in the original space are preserved in the feature space. For the neighbourhood around the equilibrium point $s_*$, there exist class $\mathcal{K}$ functions $\alpha_1$ and $\alpha_2$ as

$$\alpha_1(\|s_t - s_*\|) \leq \|\phi(s_t) - \phi(s_*)\|_2 \leq \alpha_2(\|s_t - s_*\|), \quad \forall s_t \in B(s_*, \epsilon). \tag{17}$$

$B(s_*, \epsilon)$ is denoted as $\epsilon-$ball centred at $s_*$. The smoothness of the kernel function and regularity of the dynamics ensure that the system remains well-behaved in feature space, and the convergence properties of the original system carry over to the feature space.

Thus, when the system is locally stable in the original space, the corresponding system in Hilbert space is also locally stable. According to the Hurwitz stability criterion in A.1, the linear operator $\mathcal{A}$ has only negative real part in its eigenvalues, guaranteeing exponential convergence. If $s_*$ is global equilibrium, then $\phi_S(s_*)$ is also a global equilibrium in feature space.

**Remark A.1**
*The Theorem A.3 (a) implies the existence of a global coordinate. In many situations, when the embedding space is chosen properly, we can have a stronger result that the existence of left inverse such that $\phi^\dagger(\phi(s)) = s$. This result can be naturally connected to the Kazantzis Kravaris/Luenberger (KKL) observers Tran & Bernard (2023). The embedding corresponds the injectivity of the state. The Theorem A.3 (b) corresponds the convergence in KKL observer. When the embedding space is uniformly injective, the dynamics in feature space becomes rectifiable dynamics, yielding a stable trajectory if the original system is stable.*

Before entering the last main theorem, we need to introduce the KKL observer. Based on theory of KKL observer, we will link the convergence problem in feature space with KKL observer. Please note, we assume that the state space $\mathcal{S}$ is a bounded set and system equation 1 is forward complete within this bounded set $\mathcal{S}$.

Consider the nonlinear dynamical system in equation 1 with time derivative as

$$\frac{ds_t}{dt} = f(s_t); \qquad o_t = G(s_t). \tag{18}$$

The design of KKL observer is as follows:

- Find an embedding map $\mathcal{T} : \mathcal{S} \to \mathbb{R}^{d_s}$ that transforms equation 18 to new coordinates $\mathcal{T}(s)$ as

$$\frac{\partial \mathcal{T}}{\partial s}(s)f(s) = \mathcal{A}\mathcal{T}(s) + BG(s), \tag{19}$$

where $\mathcal{A} \in \mathbb{R}^{d_s \times d_s}$ is Hurwitz matrix and $B \in \mathcal{R}^{d_s \times n_o}$, such that the system $(\mathcal{A}, B)$ is controllable[4].

- Since $\mathcal{T}$ is injective, the left inverse $\mathcal{T}^\dagger$ exists, i.e., $\mathcal{T}^\dagger(\mathcal{T}(s)) = s$. The KKL observer is then given by

$$\hat{s} = \mathcal{T}^\dagger(\mathcal{T}(\hat{s})). \tag{20}$$

There are certain conditions Andrieu & Praly (2006) that equation 18 needs to satisfy in order to ensure the existence of a KKL observer equation 19 in the sense that $\lim_{t\to\infty} \|\hat{s}_t - s_t\| = 0$. On the other hand, the map $\mathcal{T} : S \to \mathbb{R}^{d_s}$ need to be *uniformly injective* if there exists a class $\mathcal{K}$ function $\alpha$ in A.1 such that, for every $s_1, s_2 \in S$, satisfying $\|s_1 - s_2\| \leq \alpha(\mathcal{T}(s_1) - \mathcal{T}(s_2))$. Our embedding properties and convergence conditions (as shown in A.3) are satisfied the two conditions, thus we can assert the existence of KKL observer.

In this paper, we state the existence of a global linear dynamical system in feature space. We provide a theoretical guarantee that the embedding property of $\phi_S$ can derive the equivalent convergence

---

[4] $R = [\mathcal{A}, \mathcal{A}B, \mathcal{A}^2 B, ..., \mathcal{A}^{d_s-1}B]$ has full row rank (i.e. rank$(R) = d_s$)

in the feature space. However, there is two parts that have not been proven: *1). Does the global linear dynamical system exist? 2) Is the embedding space in A.3 properly defined?* We consider the nonlinear system in equation 1 within a compact space $S$ and the cost function in 2 has a unique solution. According to the condition of KKL observer, we guarantee (1) the existence of such linear system and (2) the solutions in the original space and feature space has consistent convergence properties, with respect to the cost functions equation 2 and 5, and convergent exponentially to the unique solution.

**Theorem A.4**
*Let $S$ be a bounded set in $\mathbb{R}^{n_s}$. If there exists a $C^1$ function $\mathcal{T} : S \to \mathbb{R}^{d_s}$ which satisfies the following two conditions:*

- *$\mathcal{T}$ is solution of the partial differential equation*

$$\frac{\partial \mathcal{T}}{\partial s}(s)f(s) = \mathcal{A}\mathcal{T}(s) + BG(s), \quad \forall s \in S, \tag{21}$$

  *where $\mathcal{A}$ is Hurwitz matrix, and $(\mathcal{A}, B)$ is controllable;*

- *There exists a Lipschitz constant such that for all $(s_1, s_2)$ in $S \times S$, the following inequality holds:*

$$|s_1 - s_2| \le L|\mathcal{T}(s_1) - \mathcal{T}(s_2)|; \tag{22}$$

*then there exists a continuous function $\mathcal{T}^\dagger : \mathbb{R}^{n_s} \to S$ such for all $(s, \mathcal{T}(s)) \in S \times \mathbb{R}^{n_s}$*

$$\|\mathcal{T}^\dagger(\mathcal{T}(\hat{s}_t)) - s_t\| \le cL\|\mathcal{T}(\hat{s}_0) - \mathcal{T}(s_0)\| \exp(\sigma_{max}(\mathcal{A})t), \forall t \in [0, \infty), \tag{23}$$

*and where $(s_t, \mathcal{T}(s_t))$ is the solution of system in equation 1 and equation 18 at time $t$; $\sigma_{max}(\mathcal{A})$ is the largest eigenvalue of matrix $\mathcal{A}$.*

*Proof.*

It is possible to defined the left inverse function $\mathcal{T}^\dagger : \mathbb{R}^{d_s} \to S$ and this one satisfies,

$$\|s_1 - s_2\| \le L\|\mathcal{T}(s_1) - \mathcal{T}(s_2)\|, \quad \forall(s_1, s_2) \in S \times S. \tag{24}$$

It yields that the function $\mathcal{T}^{-1} : \mathbb{R}^{d_s} \to S$ is global Lipschitz. Hence, the function $\mathcal{T}^\dagger : \mathbb{R}^{d_s} \to S$ to our problem is Lispchitz extension on the set $S$ of this function. For more convenience, we denoted $z := \mathcal{T}(s)$. Following the approach - the Mc-Shane formula in McShane (1934), we select the $\mathcal{T}^\dagger$ as the function defined by

$$\mathcal{T}^\dagger(w) \in \inf_z \left\{ (\mathcal{T}^{-1}(z) + L\|z - w\|) \right\}. \tag{25}$$

The function is such that for all $s \in S$,

$$\mathcal{T}^\dagger(\mathcal{T}(s)) = s, \tag{26}$$

and for all $(w, s) \in \mathbb{R}^{d_s} \times S$,

$$\|\mathcal{T}^\dagger(w) - s\| \le \sqrt{d_s}L\|w - \mathcal{T}(s)\|. \tag{27}$$

This implies that along the trajectory $(s_t, z_t)$ of the system satisfying the following result

$$\|\mathcal{T}^\dagger(z_t) - s_t\| \le \sqrt{d_s}L\|z_t - \mathcal{T}(s_t)\|, \quad \forall t \in [0, \infty). \tag{28}$$

On the another hand, the function $\mathcal{T}$ is solution of the partial differential equation in equation 19, consequently, this implies that along the trajectory of system $(z_t, s_t) \in \mathbb{R}^{d_s} \times S$, we have

$$z_t - \mathcal{T}(s_t) = \exp(\mathcal{A}t)(z_0 - \mathcal{T}(s_0)) \quad \forall t \in [0, \infty). \tag{29}$$

Note, since $\mathcal{A}$ is Hurwitz matrix, with $\sigma_{max}(\mathcal{A}) < 0$, it can derive that equation equation 23 holds and concludes the proof of the theorem.

**Remark A.2**
*The KKL observer asserts that the linear representation of the nonlinear system. After establishing the KKL observer, theorem A.4 asserts the convergence of the estimate trajectory to equilibrium trajectory. Since the embedding property, we can derive that the existence of left inverse based on the Mc-Shane formula in equation 25. Meanwhile, the convergence holds when pulling back the state to original space $S$. Thus we can assert the feature $\phi_S$ in our framework aligns the KKL observer, and the coordinate transformation $\mathcal{T}$ is just our feature $\phi_S$.*

## B  ESTIMATION OF ERROR MATRICES AND PSEUDO ALGORITHM

### B.1  ESTIMATION OF ERROR MATRICES.

To estimate these error covariance operators in the feature space, we empirically estimate these error matrices in the feature spaces from training dataset (Gejadze et al., 2008). For $\mathcal{R}$ and $\mathcal{Q}$, we estimated the covariances as $\mathcal{R} = \frac{1}{N}\sum_{i=1}^{N} r_i \otimes r_i$ and $\mathcal{Q} = \frac{1}{N}\sum_{i=1}^{N} q_i \otimes q_i$, where $r_i = \phi_S(s_i) - \hat{\mathcal{C}}_{S|OH}\phi_{OH}(o_i, h_i)$ and $q_i = \phi_S(s_i^+) - \hat{\mathcal{C}}_{S^+|S}\phi_S(s_i)$ are the regression residuals, quantifying the errors of the empirical operators $\hat{\mathcal{C}}_{S|OH}$ and $\hat{\mathcal{C}}_{S^+|S}$. Similarly, we compute the background covariance as the empirical variance over an average of $\{\phi_S(s_i)\}_{i=1}^{N}$[5] This error should decay monotonically over time and stabilize after a sufficiently long time horizon. This is strongly related to the covariance estimation in Kalman filtering (see Chapter 6.7 (Asch et al., 2016)). We leave the investigation of such design for future efforts.

### B.2  PSEUDO ALGORITHM.

In this section, we provide the pseudo-algorithm for training Tensor-Var with traditional kernel features in algorithm 2 and training with deep features in algorithm 1. The kernel feature map $\phi : \mathcal{S} \to \mathbb{H}_S$ such that $k(s_i, s_j) = \langle \phi(s_i), \phi(s_j)\rangle_{\mathbb{H}_S}$ may not necessarily have an explicit form (e.g., RBF and Matérn kernels), as long as the $\langle \cdot, \cdot \rangle_{\mathbb{H}_S}$ is an valid inner product. For clarity, we use the polynomial kernel with degree two and constant $c$ as an example:

- Explicit form, $\phi(s) = k(s, \cdot) = (s_1^2, ..., s_{n_s}^2, \sqrt{2}s_1 s_2, ..., \sqrt{2}s_{n_s-1}s_{n_s}, c)$
- Inner product, $k(s_i, s_j) = \langle \phi(s_i), \phi(s_j)\rangle = (s_i^T s_j + c)^2$

Algorithm 3 outlines procedure of performing data assimilation with trained models.

---

**Algorithm 1** Tensor-Var training with deep feature

**Require:** Data $\mathcal{D} = \{s_i, o_i, h_i, s_i^+\}_{i=1}^{N}$; Initialized deep features $\phi_{\theta_S}, \phi_{\theta_O}, \phi_{\theta_H}$; the inverse feature $\phi_{\theta_S'}^{\dagger}$ training epoch $K$, learning rate $\alpha$, batch size $N_B$

  **for** $k = 1, ..., K$ **do**
    Random sample batch data $\mathcal{D}_{\text{batch}} \subset \mathcal{D}$
    $\hat{\mathcal{C}}_{S^+|S}, \hat{\mathcal{C}}_{S|OH}$ = Algorithm 2 by using batch data $\mathcal{D}_{\text{batch}}$ and deep features
    Compute loss $l(\theta_S) = \|\hat{\mathcal{C}}_{S^+|S}\phi_{\theta_S}(s) - \phi_{\theta_S}(s^+)\|^2$
    Compute loss $l(\theta_O, \theta_H) = \|\hat{\mathcal{C}}_{S|OH}[\phi_{\theta_O}(o) \otimes \phi_{\theta_H}(h)] - \phi_{\theta_S}(s)\|^2$
    Compute loss $l(\theta_S, \theta_S') = \|\phi_{\theta_S'}^{\dagger}(\phi_{\theta_S}(s)) - s\|^2$
    Update the deep features.
    $\theta_S = \theta_S + \alpha\nabla_{\theta_S}l(\theta_S)$;
    $\theta_O, \theta_H = \theta_O + \alpha\nabla_{\theta_O}l(\theta_O, \theta_H), \theta_H + \alpha\nabla_{\theta_H}l(\theta_O, \theta_H)$;
    $\theta_S, \theta_S' = \theta_S + \alpha\nabla_{\theta_S}l(\theta_S, \theta_S'), \theta_S' + \alpha\nabla_{\theta_S'}l(\theta_S, \theta_S')$
  **end for**
  Compute $\hat{\mathcal{C}}_{S^+|S}, \hat{\mathcal{C}}_{S|OH}$ = Algorithm 2 by using the whole dataset $\mathcal{D}$ and trained deep features $\phi_{\theta_S}, \phi_{\theta_O}, \phi_{\theta_H}$.
  Compute the error covariance matrices $\mathcal{B}, \mathcal{R}, \mathcal{Q}$ from subsection B.1
  **return** $\phi_{\theta_S}, \phi_{\theta_O}, \phi_{\theta_H}, \hat{\mathcal{C}}_{S^+|S}, \hat{\mathcal{C}}_{S|OH}, \mathcal{B}, \mathcal{R}$, and $\mathcal{Q}$

---

[5]For a cyclic application of Tensor-Var, a better design for $\mathcal{B}$ should be time-dependent, reflecting the error between the estimated system state and the true state, e.g. (Paulin et al., 2022).

---

**Algorithm 2** Tensor-Var training with kernel feature

---

**Require:** Dataset $\mathcal{D} = \{s_i, o_i, h_i, s_i^+\}_{i=1}^N$; kernel features $\phi_S(s) = k_S(s, \cdot), \phi_O(o) = k_O(o, \cdot), \phi_H(h) = k_H(h, \cdot)$
  Compute the Gram matrix $K_S$ where $[K_S]_{ij} = k_S(s_i, s_j)$
  Compute the Gram matrix $K_{OH}$ where $[K_{OH}]_{ij} = k_{OH}(o_i \otimes h_i, o_j \otimes h_j) = k_O(o_i, o_j)k_H(h_i, h_j)$

  If $N$ is too large, say $N \geq 10000$, using the Nystrom approximation to select a subset $\mathcal{D}^s = \{s_i, o_i, h_i, s_i^+\}_{i=1}^n$
  Compute the feature matrix $\Phi_S = [\phi_S(s_1), ..., \phi_S(s_n)]$
  Compute the feature matrix $\Phi_{S^+} = [\phi_S(s_1^+), ..., \phi_S(s_n^+)]$
  Compute the feature matrix $\Phi_{OH} = [\phi_{O,H}(o_1, h_1), ..., \phi_{O,H}(o_n, h_n)]$
  CME for the system dynamics. $\hat{\mathcal{C}}_{S^+|S} = \Phi_{S^+}(K_S + \lambda I)^{-1}\Phi_S$
  CME for the inverse observation model. $\hat{\mathcal{C}}_{S|OH} = \Phi_S(K_{OH} + \lambda I)^{-1}\Phi_{OH}^T$
  Compute the error covariance matrices $\mathcal{B}, \mathcal{R}, \mathcal{Q}$ from subsection B.1.
  Fit the projection matrix for pre-image. $\hat{\mathcal{C}}_{\text{proj}} = \mathbf{S}(K_S + \lambda I)^{-1}\Phi_S^T$ where $\mathbf{S} = (s_1, ..., s_n)$
  **return** $\hat{\mathcal{C}}_{S^+|S}, \hat{\mathcal{C}}_{S|OH}, \hat{\mathcal{C}}_{\text{proj}}, \mathcal{B}, \mathcal{R},$ and $\mathcal{Q}$

---

---

**Algorithm 3** Tensor-Var assimilation-forecasting

---

**Require:** assimilation window $\{o_t, h_t\}_{t=0}^T$; background state $s_b$; leading time $\tau$; kernel features $\phi_S, \phi_O, \phi_H$ (or trained deep features $\phi_{\theta_S}, \phi_{\theta_O}, \phi_{\theta_H}$); CME operators $\hat{\mathcal{C}}_{S^+|S}$ and $\hat{\mathcal{C}}_{S|OH}$; Error covariance matrices $\mathcal{B}, \mathcal{R}, \mathcal{Q}$.
  Perform Quadratic Programming with objective

$$\min_{\{z_t\}_{t=0}^T} \|z_0 - \phi_S(s_0^b)\|_{\mathcal{B}^{-1}}^2 + \sum_{t=0}^T \|z_t - \hat{\mathcal{C}}_{S|OH}\phi_{OH}(o_t, h_t)\|_{\mathcal{R}^{-1}}^2$$

$$+ \sum_{t=0}^{T-1} \|z_{t+1} - \hat{\mathcal{C}}_{S^+|S}z_t\|_{\mathcal{Q}^{-1}}^2,$$

  Project back to original space with $\hat{s}_t = \hat{\mathcal{C}}_{\text{proj}}z_t$ (or using learned inverse feature $\hat{s}_t = \phi_{\theta_S'}^\dagger(z_t)$)
  **for** $t = 1, ..., \tau$ **do**
    Predict $z_{t+T} = \hat{\mathcal{C}}_{S^+|S}z_{T+t-1}$
    Project back to original space with $\hat{s}_t = \hat{\mathcal{C}}_{\text{proj}}z_t$ (or inverse feature $\hat{s}_{T+t} = \phi_{\theta_S'}^\dagger(z_{T+t})$)
  **end for**

---

## C  EXPERIMENT SETTINGS

**Training details.** Given the generated data, we constructed two datasets: $\mathcal{D}_{\text{dyn}} = \{\{(s_t^i, s_{t+1}^i)\}_{t=0}^{T-1}\}_{i=1}^N$ and $\mathcal{D}_{\text{obs}} = \{\{(s_t^i, o_t^i, h_t^i)\}_{t=0}^T\}_{i=1}^N$. The DFs were trained in two steps using these datasets. First, the state DFs $\phi_{\theta_S}, \phi_{\theta_S'}^\dagger$ were trained on $\mathcal{D}_{\text{dyn}}$ using equation 6 and we stored the estimated operator $\hat{\mathcal{C}}_{S+|S}$. Next, with the state features fixed, the observation DF $\phi_{\theta_O}$ and history DF $\phi_{\theta_H}$ were trained on $\mathcal{D}_{\text{obs}}$ according to equation 3.2, storing the estimated operator $\hat{\mathcal{C}}_{S|OH}$. The baseline method (Frerix et al., 2021) was trained on $\mathcal{D}_{\text{obs}}$, excluding history. All models were trained with the Adam optimizer (Kingma, 2014) for 200 epochs, using batch sizes from 256 to 1024 for stable operator estimation. Additional details on the DFs, baselines, and training procedures can be found in Appendix C.

**Implementation details.** For all baseline methods, we employed the L-BFGS algorithm for Variational Data Assimilation (Var-DA) optimization, implemented in JAX (Bradbury et al., 2018). The 4D-Var baselines used numerical dynamical models based on the 8th-order Runge-Kutta method and the 4th-order Exponential Time Differencing Runge-Kutta (ETDRK) method (Cox & Matthews, 2002) for the Lorenz-96 and KS systems. For Tensor-Var, we applied interior-point quadratic programming to solve the linearized 4D-Var optimization, utilizing CVXPY (Diamond & Boyd, 2016). All training was conducted on a workstation with a 48-core AMD 7980X CPU and an Nvidia GeForce 4090 GPU. Runtime evaluations were performed on a MacBook with an 8-core Apple M1 Pro CPU, without GPU acceleration.

### C.1  LORENZ 96

First, we consider the single-level Lorenz-96 system, which was introduced in (Lorenz, 1996) as a low-order model of atmospheric circulation along a latitude circle. The system state is $[S_1, ..., S_K]$ representing atmospheric velocity at $K$ evenly spaced locations and is evolved according to the governing equation:

$$\frac{dS_k}{dt} = -S_{k-1}(S_{k-2} - S_{k+1}) - S_k + F,$$

with periodic boundary conditions $x_{k+K} = x_k$. The first term models advection, and the second term represents a linear damping with magnitude $F$. In general, the dynamics becomes more turbulent/chaotic as F increases. We choose the number of variables $K = 40, 80$ and the external forcing $F = 10$, where the system is chaotic with a Lyapunov time of approximately 0.6 time units. As an observation model for the following experiments, we randomly observe 25% states (e.g. 10 in $K = 40$). Our models were trained on a dataset $\mathcal{D}$ of $N = 100$ trajectories, each trajectory consist of $= 5000$ time steps long, generated by integrating the system from randomly sampled initial conditions.

#### C.1.1  DATA GENERATION.

To generate the dataset, we use the 8th-order Runge-Kutta (Butcher, 1996) method to numerically integrate the Lorenz-96 systems with sample step 0.1 and the integration step $\Delta t$ size is set to 0.01. The system is integrated from randomly sampled initial conditions, and data is collected once the system reaches a stationary distribution. For an observation operator, we use subsampling which every 5th and 10th variable for 40 and 80-dimensional system are observed via the nonlinear mapping $5\arctan(\cdot\pi/10) + \epsilon$ with noise $\epsilon \sim \mathcal{N}(0, 0.1)$ (see Figure 7 in Appendix C.1 for an example). The $\arctan\colon \mathbb{R} \mapsto [-\frac{\pi}{2}, \frac{\pi}{2}]$ squeezes the state variable $S_k$ into $[-\frac{\pi}{2}, \frac{\pi}{2}]$, which is difficult for inverse estimation. We integrate the Lorenz96 system with observation interval $\Delta t = 0.1$. The history length is set as 10 such that $h_t = (o_{t-10}, ..., o_{t-1})$.

#### C.1.2  ADDITIONAL EXPERIMENT RESULTS.

We provide qualitative results in Figure 8 for the Lorenz 96 system at two different dimensions: 40 (left) and 80 (right). Each subplot illustrates the normalized absolute error for various methods, including 3D-VAR, 4D-VAR, Frerix et al. (2021), and Tensor-Var, compared to the ground truth. The assimilation window length is set to 5 (indicated by the red dashed line), with forecasts extended for an additional 100 steps based on the assimilated results. Tensor-Var generally outperforms the

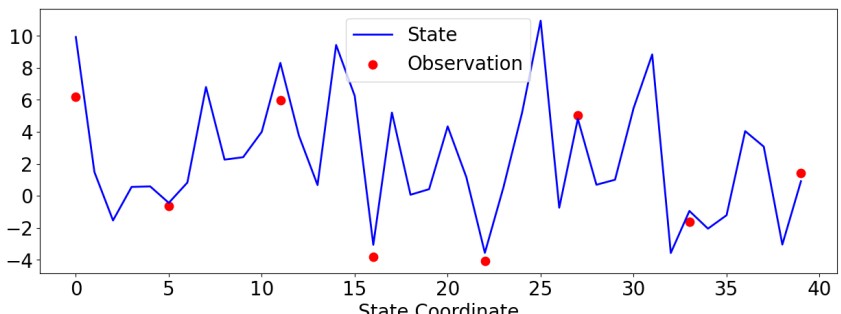

Figure 7: Nonlinear observation model in the Lorenz 96 system: the state values are represented by the solid blue curve, with the observed grid points indicated by red dots.

other methods in both dimensions and maintains stable long-term forecasts, comparable to other model-based approaches. For 3D- and 4D-VAR with partially observed models, the observed states show minimized errors (indicated by the dark lines), while the errors of unobserved states remain uncontrollable, as clearly shown in Figure 8.

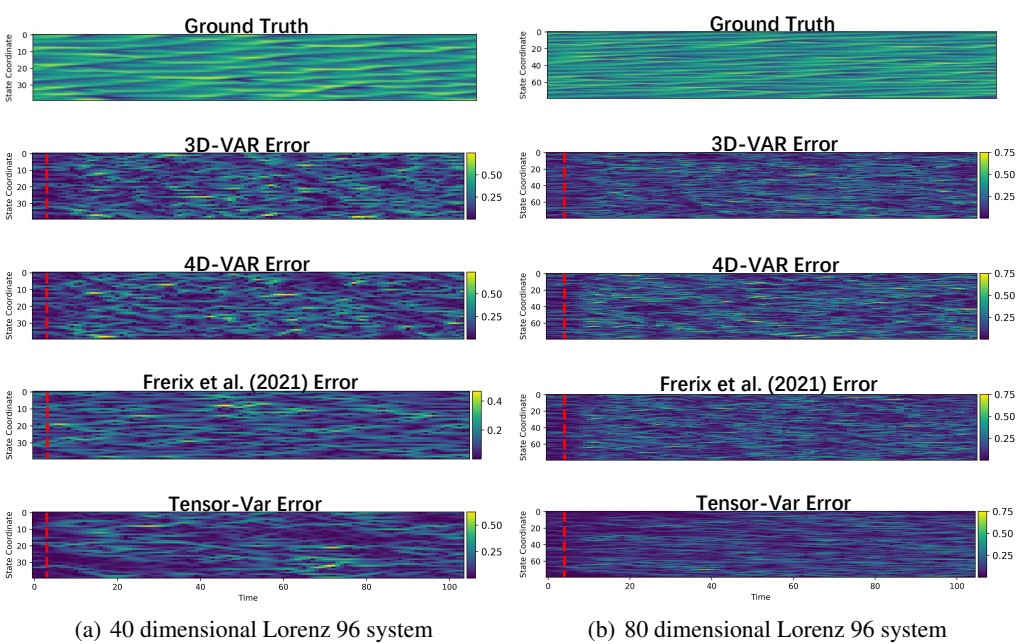

(a) 40 dimensional Lorenz 96 system      (b) 80 dimensional Lorenz 96 system

Figure 8: Qualitative error comparison for the Lorenz 96 system at (a) 40 dimensions and (b) 80 dimensions. The plots show the normalized absolute errors for various methods, including 3D-VAR, 4D-VAR, Frerix et al. (2021), and Tensor-Var, compared to the ground truth. The assimilation window length is set to 5 (indicated by the red dashed line), with forecasts extended for an additional 100 steps based on the assimilated results.

## C.2 KURAMOTO-SIVASHINSKY

Next, we consider the Kuramoto-Sivashinsky (KS) equation, a nonlinear PDE system known for its chaotic behaviour and widely used to study instability in fluid dynamics and plasma physics (Papageorgiou & Smyrlis, 1991). The dynamics in spatial domain $u(x, t)$ is given by,

$$\frac{\partial u}{\partial t} + u\frac{\partial u}{\partial x} + \frac{\partial^2 u}{\partial x^2} + \frac{\partial^4 u}{\partial x^4} = 0,$$

where $x \in [0, L]$ with periodic boundary conditions. We set the domain length $L = 32\pi$, large enough to induce complex patterns and temporal chaos due to high-order term interactions (Cvitanović et al., 2010). The system state $u(x, t)$ was discretized into $n_s = 128$ and $n_s = 256$. The observation model is the same as in Lorenz-96, where $25\%$ states can be observed. In this case, our models were trained on a dataset $\mathcal{D}$ consisting of $N = 100$ trajectories, each with $L = 5000$ time steps and a discretization of $\Delta t = 0.01$, sampled from the stationary distribution with different initial conditions.

### C.2.1 DATA GENERATION.

To generate the dataset, we use the exponential time differencing Runge–Kutta method (ETDRK), which has proven effective in computing nonlinear partial differential equation (Cox & Matthews, 2002) with an integration step $\Delta t = 0.001$ and sample step $0.01$. The system is integrated from randomly sampled initial conditions, and data is collected once it reaches a stationary distribution. For observations, we use subsampling, observing every 8th state in both 128- and 256-dimensional systems, we use subsampling which every 8th for both 128 and 256-dimensional system are observed with noise $\epsilon \sim \mathcal{N}(0, 1)$, and $5 \arctan(\cdot\pi/10)$ as nonlinear mapping (see Figure 7 in Appendix C.1 for an example). The history length is set to 10 as well.

### C.2.2 ADDITIONAL EXPERIMENT RESULTS.

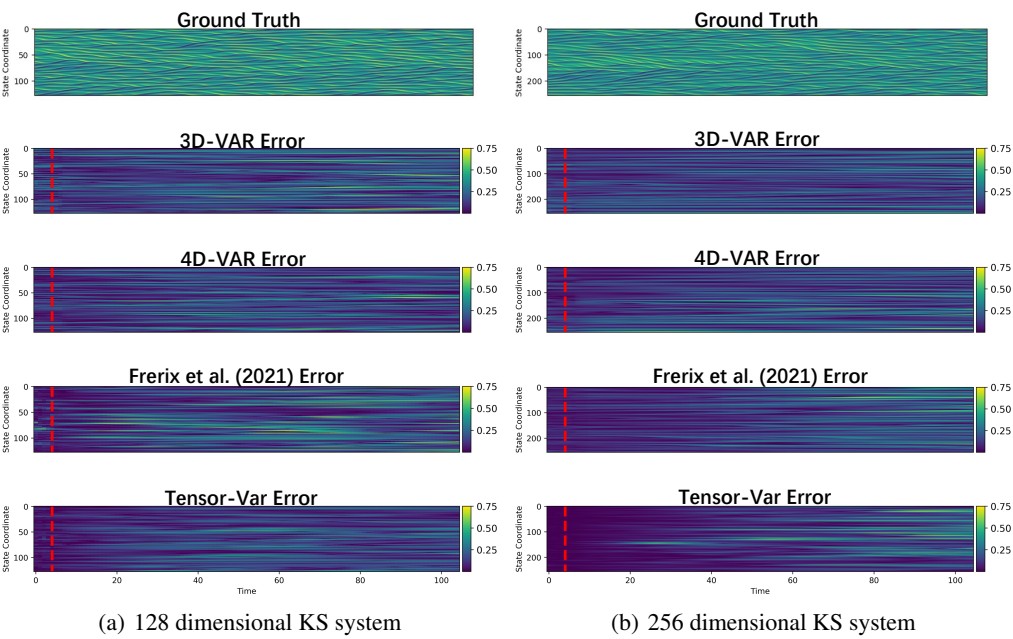

(a) 128 dimensional KS system          (b) 256 dimensional KS system

Figure 9: Qualitative error comparison for the KS system at (a) 128 dimensions and (b) 256 dimensions. The plots show the normalized absolute errors for various methods, including 3D-VAR, 4D-VAR, Frerix et al. (2021), and Tensor-Var, compared to the ground truth. The assimilation window length is set to 5 (indicated by the red dashed line), with forecasts extended for an additional 100 steps based on the assimilated results.

We provide qualitative results in Figure 9 for the KS systems at two different dimensions: 128 (left) and 256 (right). Each subplot illustrates the normalized absolute error for various methods,

including 3D-VAR, 4D-VAR, Frerix et al. (2021), and Tensor-Var, compared to the ground truth. The assimilation window length is set to 5 (indicated by the red dashed line), with forecasts extended for an additional 100 steps based on the assimilated results.

Compared to the Lorenz-96 system, the KS system is more complex, being governed by partial differential equations (PDEs) that account for spatial evolution. In both dimensions, Tensor-Var consistently outperforms other methods, particularly in capturing chaotic dynamics during the initial forecast phase. It also maintains long-term stability in a more complex PDE system, comparable to other model-based approaches. In contrast, 3D-VAR struggles with assimilation, especially in the 256-dimensional case, due to its inability to capture temporal evolution, leading to rapid error divergence. This underscores the critical importance of temporal modeling in chaotic systems. A similar pattern of error between observed and unobserved states is evident in Figure 9.

## C.3 GLOBAL NWP

We consider a global numerical weather prediction (NWP) problem using the European Centre for Medium-Range Weather Forecasts (ECMWF) Atmospheric Reanalysis (ERA5) dataset (Hersbach et al., 2020). This dataset provides high-resolution atmospheric reanalysis data from 1940 to the present, offering the most comprehensive estimate of atmospheric dynamics. For our proof of concept, we focus on five upper level physical variables: 500 hPa geopotential height, 850 hPa temperature, 700 hPa humidity, and 850 hPa wind speed (meridional and zonal components) at a spatial resolution of 64×32.

The data is sourced from the WeatherBench2 repository (Rasp et al., 2024). From this dataset, we randomly sample grid points with 15% spatial coverage. The sampled observations include additive noise equivalent to 0.01 times the standard deviation of the state variable, ensuring robustness against observational uncertainty (see Figure 2). For model training, we use ERA5 data from 1979-01-01 to 2016-01-01, separating data from post-2018 for testing. There were 51,100 consecutive system states with generated observations for training and 2,920 data for testing.

In addition to the results presented in the main experiments, we evaluate the forecasting quality of Tensor-Var based on the assimilated state. Figure 10 shows the mean latitude-weighted RMSE (Rasp et al., 2024) for five variables predicted by Tensor-Var at various lead times $\tau$, where $\tau = 0$ represents the assimilation error at the final state of the assimilation window.

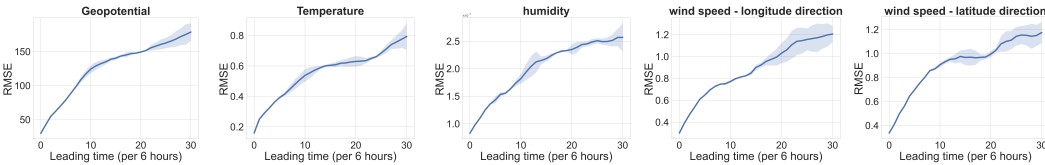

Figure 10: The (non-cyclic) forecasting quality of Tensor-Var in NWP experiments with leading time zero as the final state in the assimilation window, is evaluated across different experiments. The five sub-figures display the NWP forecast for 5 variables (15-day in total).

**Area-weighting Root mean squared error (RMSE).**   The error is defined for each variable and level as

$$\sqrt{\frac{1}{TIJ} \sum_{t=1}^{T} \sum_{i=1}^{I} \sum_{j=1}^{J} (w(i)\hat{s}_{t,i,j} - s_{t,i,j})^2},$$

which is area-weighting over grid points. This is because on an equiangular latitude-longitude grid, grid cells at the poles have a much smaller area compared to grid cells at the equator. Weighing all cells equally would result in an inordinate bias towards the polar regions. The latitude weights $w(i)$ are computed as:

$$w(i) = \frac{\sin\left(\theta_i^u\right) - \sin\left(\theta_i^l\right)}{\frac{1}{I} \sum_{i=1}^{I} (\sin\left(\theta_i^u\right) - \sin\left(\theta_i^l\right))},$$

$\theta_u^i$ and $\theta_l^i$ indicate upper and lower latitude bounds, respectively, for grid cell with latitude index $i$.

## C.4 ASSIMILATION FROM SATELLITE OBSERVATIONS

### C.4.1 DATA GENERATION.

We collected the weather satellite track data from `https://celestrak.org/NORAD/elements/` for the period 1979-01-01 00:00:00 to 2020-01-01 00:00:00. Observations were matched to the high-resolution ERA5 dataset (240 × 121 grid) by identifying the nearest neighborhood grid points along the satellite track to generate observations. Additionally, we used an observation frequency of up to every half-hour within the 2 hours before each assimilation time and added white noise with a standard deviation of 1% of the standard deviation of the corresponding state variables.

### C.4.2 ADDITIONAL QUALITATIVE RESULTS

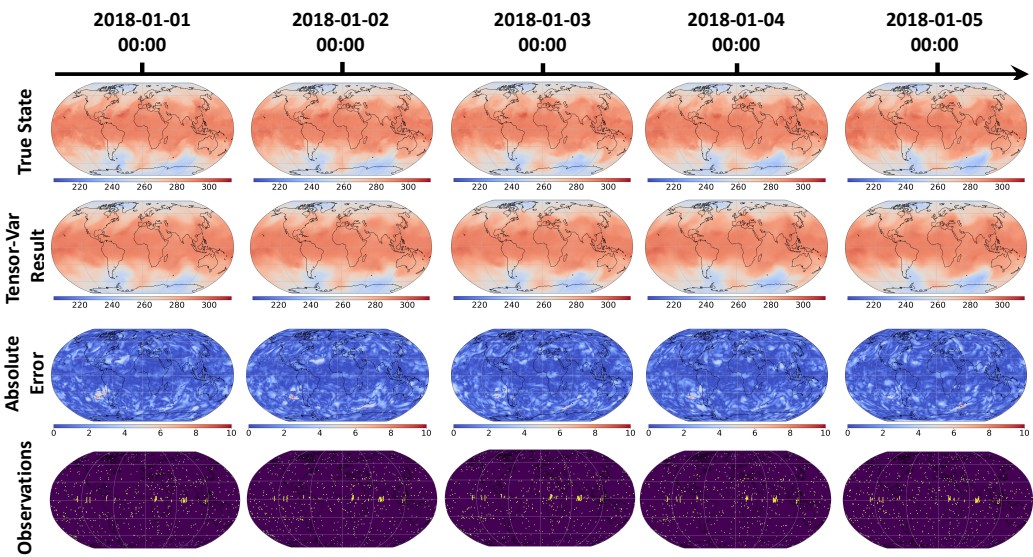

Figure 11: Visualization of continuous assimilation results, absolute errors, and observation locations for t850 (temperature), starting from 2018-01-01 00:00.

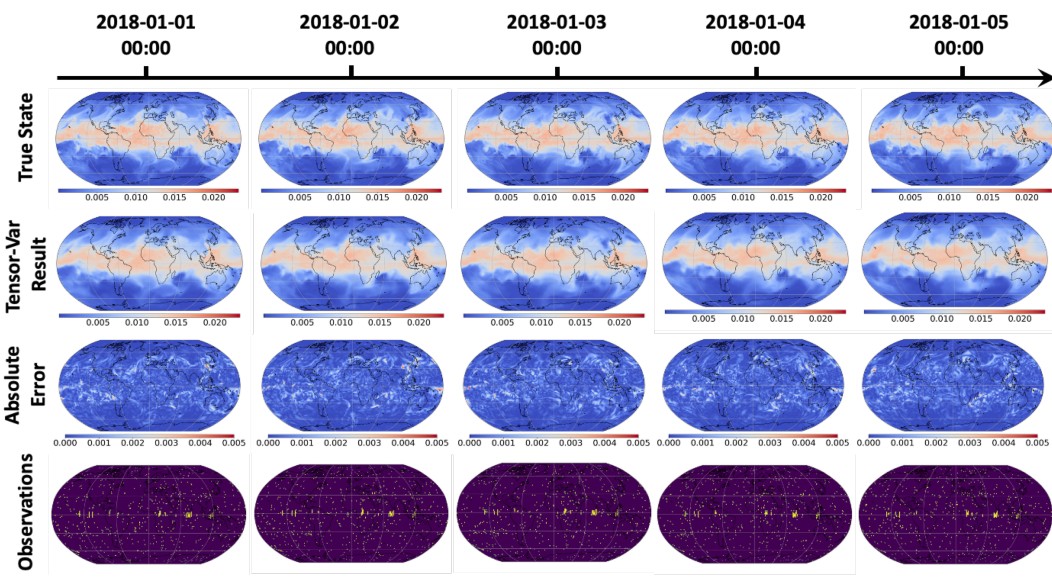

Figure 12: Visualization of continuous assimilation results, absolute errors, and observation locations for q700 (humidity), starting from 2018-01-01 00:00.

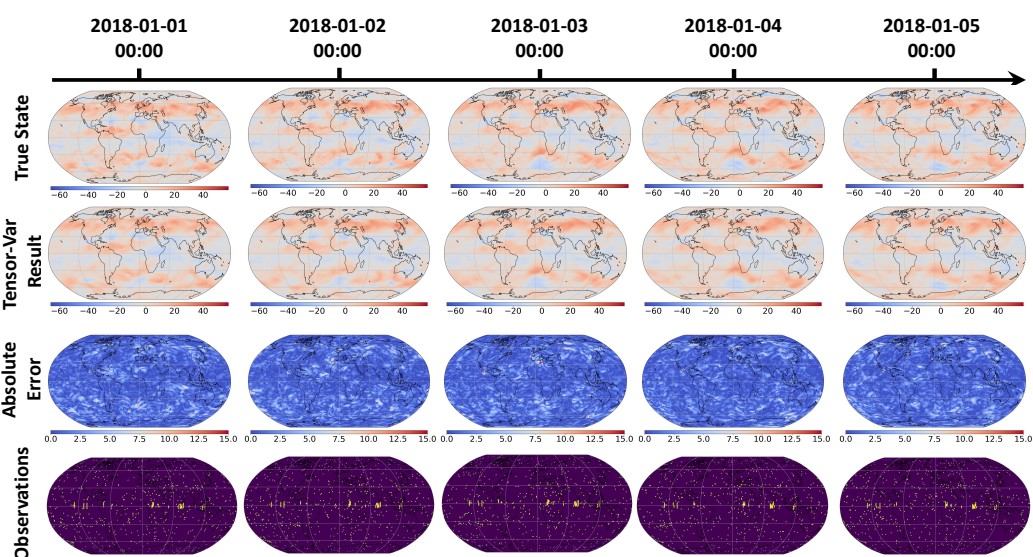

Figure 13: Visualization of continuous assimilation results, absolute errors, and observation locations for u850 (meridional wind speed), starting from 2018-01-01 00:00.

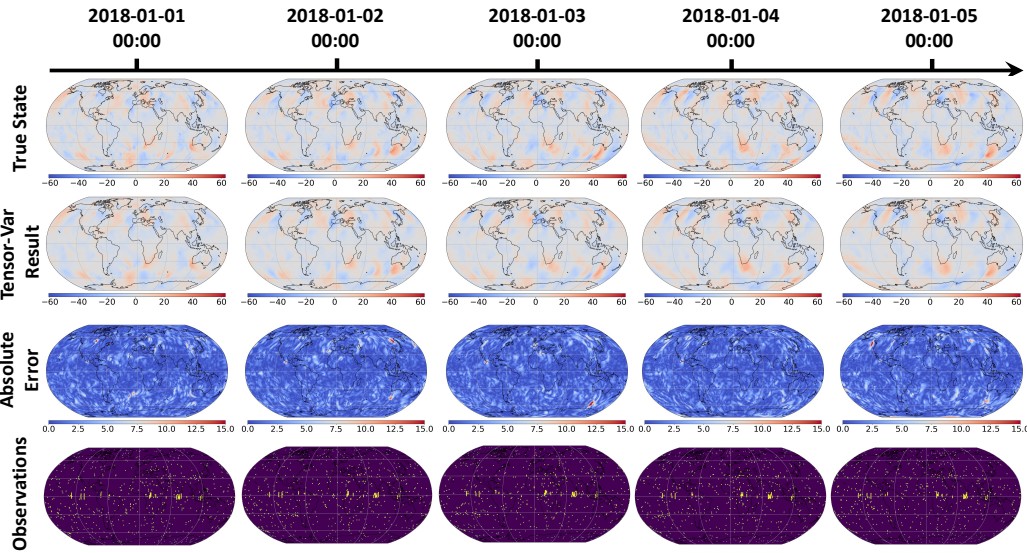

Figure 14: Visualization of continuous assimilation results, absolute errors, and observation locations for v850 (zonal wind speed), starting from 2018-01-01 00:00.

C.5 ABLATION STUDY

In this section, we provide the details of the ablation studies of (1) the history length $m$, (2) the dimensions of feature dimension and comparison with standard kernel functions, and (3) the effects of the estimated error matrices. All the ablation studies are conducted on the Lorenz-96 systems with $n_s = 40$ and $n_s = 80$. We fixed the remaining hyperparameters consistent with the main experiments and varied only the parameters under investigation.

**(1) Effect of the history length** $m$.    We examined the impact of the history length $m$ on learning the inverse observation operator and its effect on state estimation accuracy. The feature dimensions $d_s$, $d_o$, and $d_h$ were held constant, while the history length was varied by adjusting the size of the final linear layer (see Table 4). The state feature dimensions $d_s$ were set to 60 and 120 for the two system dimensions.

**(2) Effect of DFs.**    We implemented Tensor-Var with a Gaussian kernel using kernel PCA projected to first 60 and 120 eigen-coordinates in scikit-learn (Pedregosa et al., 2011) by aligning with the dimension of the used DFs. For the Gaussian kernel, we approach the pre-image problem by fitting a projection operator (Kwok & Tsang, 2004). The space $\mathcal{S} \subset \mathbb{R}^{n_s}$ together with the linear kernel $k(s_i, s_j) = s_i^T s_j$ forms an RKHS as well. Therefore, we can simply define the projection operator as a CME operator that maps from the feature space $\mathbb{H}_S$ to the original space $\mathcal{S}$ as $\hat{\mathcal{C}}_{\text{proj}} = \mathbf{S}(K_S + \lambda I)^{-1} \Phi_S^T$, where $\mathbf{S} = [s_1, ..., s_n]$. By applying $\hat{\mathcal{C}}_{\text{proj}}$ now to the state, we can obtain the mean estimation of the kernel mean embedding in the $\mathcal{S}$ such that $\hat{s} = \hat{\mathcal{C}}_{\text{proj}} \mu_{\mathbb{P}_S} = \mathbb{E}_S[\hat{\mathcal{C}}_{\text{proj}} \phi_S(S)] = \mathbb{E}_S[S]$. We normalized the dataset to a standard Gaussian distribution and select the length scale $\gamma = 1.0$ by performing a cross-validation on the $\gamma = [0.5, 0.75, 1.0, 1.25, 1.5, 2]$.

## C.6 MODEL ARCHITECTURE

Table 4: Deep feature architecture for 1D chaotic systems with dimensions $n_s, n_o, n_h = m \times n_o$ and feature dimension $d_s, d_o, d_h$

| Components | Layer | Weight size | Bias size | Activation |
|---|---|---|---|---|
| $\phi_{\theta_S}$ | Fully Connected | $n_s \times 4n_s$ | $4n_s$ | Tanh |
| | Fully Connected | $4n_s \times 2n_s$ | $2n_s$ | Tanh |
| | Fully Connected | $2n_s \times d_s$ | $d_s$ | |
| $\phi_{\theta'_S}^{\dagger}$ | Fully Connected | $d_s \times 2n_s$ | $2n_s$ | Tanh |
| | Fully Connected | $2n_s \times 4n_s$ | $4n_s$ | Tanh |
| | Fully Connected | $4n_s \times n_s$ | $n_s$ | |
| $\phi_{\theta_O}$ | Fully Connected | $n_o \times 4n_o$ | $4n_o$ | Tanh |
| | Fully Connected | $4n_o \times 2n_o$ | $2n_o$ | Tanh |
| | Fully Connected | $2n_o \times d_o$ | $d_o$ | |
| $\phi_{\theta_H}$ | Convolution 1D | $m \times 2m \times 5$ | $2m$ | Tanh |
| | Max Pooling (size=2) | | | |
| | Convolution 1D | $2m \times 4m \times 3$ | $4m$ | Tanh |
| | Max Pooling (size=2) | | | |
| | Flatten | | | |
| | Fully Connected | $mn_o \times d_h$ | $d_h$ | |

Table 5: Model architecture for Global NWP with input dimension $(H, W, C)$ with $C$ physical variables and spatial resolution $H \times W$. The implementation of vision Transformer (ViT) block follows Zamir et al. (2022) with applications in DA followed by Nguyen & Fablet (2024).

| Components | Layer | Layer number | $C, (H, W)$ | Activation |
|---|---|---|---|---|
| $\phi_{\theta_S}$ | Convolution2d | 1 | $C \to 4C, (H, W)$ | |
| | Transformer Block | 2 | $4C \to 4C, (\frac{H}{2}, \frac{W}{2})$ | ReLU |
| | Transformer Block | 3 | $4C \to 8C, (\frac{H}{4}, \frac{W}{4})$ | ReLU |
| | Transformer Block | 3 | $8C \to 8C, (\frac{H}{8}, \frac{W}{8})$ | ReLU |
| | Flatten | | $(8C, \frac{H}{8}, \frac{W}{8}) \to \frac{CHW}{8}$ | |
| | Fully Connected | 1 | $\frac{CHW}{8} \to d_s$ | |
| $\phi_{\theta'_S}^{\dagger}$ | Fully Connected | 1 | $d_s \to \frac{CHW}{8}$ | |
| | Transpose | | $\frac{CHW}{8} \to (8C, \frac{H}{8}, \frac{W}{8})$ | |
| | Transformer Block | 3 | $8C \to 8C, (\frac{H}{8}, \frac{W}{8})$ | ReLU |
| | Transformer Block | 3 | $8C \to 4C, (\frac{H}{4}, \frac{W}{4})$ | ReLU |
| | Transformer Block | 2 | $4C \to 4C, (\frac{H}{2}, \frac{W}{2})$ | ReLU |
| | Convolution2d | 1 | $4C \to C, (H, W)$ | |
| $\phi_{\theta_O}$ | Convolution2d | 1 | $C \to 2C, (H, W)$ | |
| | Transformer Block | 2 | $2C \to 2C, (\frac{H}{2}, \frac{W}{2})$ | ReLU |
| | Transformer Block | 3 | $2C \to 4C, (\frac{H}{8}, \frac{W}{8})$ | ReLU |
| | Flatten | | $(4C, \frac{H}{8}, \frac{W}{8}) \to \frac{CHW}{16}$ | |
| | Fully Connected | 1 | $\frac{CHW}{16} \to d_o$ | |
| $\phi_{\theta_O}$ | Convolution2d | 1 | $mC \to 2C, (H, W)$ | |
| | Transformer Block | 2 | $2C \to 2C, (\frac{H}{2}, \frac{W}{2})$ | ReLU |
| | Transformer Block | 3 | $2C \to 4C, (\frac{H}{8}, \frac{W}{8})$ | ReLU |
| | Flatten | | $(4C, \frac{H}{8}, \frac{W}{8}) \to \frac{CHW}{16}$ | |
| | Fully Connected | 1 | $\frac{CHW}{16} \to d_h$ | |

