# OpenReview forum: "Tensor-Var: Variational Data Assimilation in Tensor Product Feature Space"
_ICLR.cc/2025/Conference — ICLR 2025 Conference Withdrawn Submission_

### Official Review · Reviewer_iTY2 · 2024-11-03

**Soundness:** 2
**Presentation:** 2
**Contribution:** 3
**Rating:** 6
**Confidence:** 3

**Summary:**

This paper utilizes the CME method to improve the efficiency of 4D-Var; Several toy scenarios are used to study the assimilation ability of the TENSOR-VAR in chaotic systems and global weather forecasting. Observations are simulated by sampling the dataset, rather than using true observations, for the assimilation of the background field. Ultimately, this work achieves a toy assimilation model with limited significance.

**Strengths:**

The article thoroughly discusses how the Conditional Mean Embedding (CME) method can further improve the efficiency of 4D-Var. It presents the mathematical principles of the method clearly, albeit with some redundancy, and employs several toy scenarios to compare various assimilation approaches.

**Weaknesses:**

1. The resolution is too low, and it seems that all observations are located at grid points (If not, please correct it), which greatly simplifies the problem.

2. Real observations are not used in assimilation:

(1) Real observations are often not randomly distributed and typically exhibit some certain patterns.
(2) The ERA5 dataset is already a reanalysis dataset (incorporating numerous observations and numerical simulations). Using random samples from ERA5 to simulate assimilation means that the model assimilates only one type of data, which differs significantly from reality. This article does not address or explain the assimilation of multiple sources of observations.
(3) A key aspect of assimilation is real-time processing (unlike data fusion), and the experimental design of this paper does not explore online operational scenarios.

In summary, I believe this work is still far from real assimilation applications. The research method and experimental design are relatively detached from real-world conditions, which significantly reduces the study's relevance and value.

In summary, I believe this work is still far from practical assimilation applications. The research methods and experimental design are relatively detached from real-world conditions, which significantly diminishes the study’s relevance and value. It creates some barriers to understanding. The article may not be particularly accessible to readers in physics-related fields, such as atmospheric science, complex systems, and general physics.

**Questions:**

In weaknesses part.

---

> ### Author Response · Authors · 2024-11-22
> **Response to Weakness**
>
> We sincerely thank the reviewer for the time and effort in reviewing our work and appreciate the valuable comments.
>
> However, we feel that certain aspects of your comments are overly critical and do not fully engage with the scientific merits of our work. The characterization of the mathematical derivations as "sophisticated but redundant" seems dismissive without considering their theoretical necessity. Our intention was not to complicate the presentation but to provide a rigorous and complete foundation, which is essential in the field of machine learning, where precision and depth are highly valued. Our mathematical formulation follows well-established conventions in kernel conditional meaning embedding through publications in the Machine Learning community, including Muandet, et al., 2017; Tolstikhin, et al., 2017; Simon-Gabriel, et al., 2018; Hayati, et al., 2024, and is carefully adapted to our problem context in accordance with the standards of ICLR.
>
> In response to the reviewer’s concern, we added a practical global numerical weather prediction (NWP) experiment using real-time weather satellite observations (available at https://celestrak.org/, data including satellite observations from ISS (ZARYA), POISK, COSMOGIRLSAT and SHENZHOU-19) with a spatial resolution of up to 240×121, compared to the 64×32 resolution used in the original submission. This includes comparisons with the Latent 3D-Var and Latent 4D-Var baselines. The results demonstrate that Tensor-Var achieves robust performance in terms of accuracy and robustness for large-scale systems with real-world observations, highlighting its potential for operational data assimilation and forecasting applications.
> The complexities of this experiment, with more realistic observations, exceeds that of similar published variational DA papers **in the machine learning community (McCabe, et al., 2021; Frerix, et al., 2021; Rozet, et al., 2023; Xiao, et al., 2024)**. Although operational data assimilation system is not the primary scope of ICLR, we believe we have provided a meaningful contribution with practical impact.
>
> Finally, we would like to emphasize that our study is positioned within the scope of ICLR, focusing on advancing methodological frameworks in representation learning, with kernel theory as a significant branch, and providing theoretical insights for Variational DA. To best of our knowledge, our work firstly introduces the kernel conditional mean embedding framework into the 4D-Var data assimilation and the brought advantages such as linearization and convex cost objective are very desired in the application of 4D-Var. We believe our work do a meaningful start and pave a way for application of kernel methods in the domain of variatioal DA.
>
> In the end, we thank the reviewer for the time and effort in reviewing our work and the provided comments on real-world applications. We hope the reviewer can have a look our updated manuscript which have revised to improve its readability in section 3 and added experiment in section 4.3. We look forward to your further comments and suggestions.
>
> - References
>
>   - Muandet, Krikamol, et al. "Kernel mean embedding of distributions: A review and beyond." Foundations and Trends® in Machine Learning 10.1-2 (2017): 1-141.
>
>   - Tolstikhin, Ilya, Bharath K. Sriperumbudur, and Krikamol Mu. "Minimax estimation of kernel mean embeddings." Journal of Machine Learning Research 18.86 (2017): 1-47.
>
>   - Simon-Gabriel, Carl-Johann, and Bernhard Schölkopf. "Kernel distribution embeddings: Universal kernels, characteristic kernels and kernel metrics on distributions." Journal of Machine Learning Research 19.44 (2018): 1-29.
>
>   - Hayati, Saeed, Kenji Fukumizu, and Afshin Parvardeh. "Kernel mean embedding of probability measures and its applications to functional data analysis." Scandinavian Journal of Statistics 51.2 (2024): 447-484.
>
>   - Frerix, Thomas, et al. "Variational data assimilation with a learned inverse observation operator." International Conference on Machine Learning. PMLR, 2021.
>
>   - Rozet, François, and Gilles Louppe. "Score-based data assimilation." Advances in Neural Information Processing Systems 36 (2023): 40521-40541.
>
>   - McCabe, Michael, and Jed Brown. "Learning to assimilate in chaotic dynamical systems." Advances in neural information processing systems 34 (2021): 12237-12250.
>
>   - Xiao, Yi, et al. "Towards a Self-contained Data-driven Global Weather Forecasting Framework." Forty-first International Conference on Machine Learning.

---

> > ### Comment · Reviewer_iTY2 · 2024-11-24
> >
> > Thank you for your detailed response. Considering your additional experiments, I have re-evaluated my review.

---

> > > ### Author Response · Authors · 2024-11-24
> > >
> > > We would like to thank reviewer iTY2 for carefully reviewing our response and additional experiments. Your feedback helped us greatly improve the paper, and we greatly appreciate your re-evaluation and updated score.

---

### Official Review · Reviewer_9XsZ · 2024-11-03

**Soundness:** 3
**Presentation:** 2
**Contribution:** 2
**Rating:** 6
**Confidence:** 3

**Summary:**

This work presents *Tensor-Var*, an efficient 4D-Var computational framework. The key component of the framework are the linear representations of the dynamical systems and state-observation mappings obtained by kernel Conditional Mean Embedding. These transform the nonlinear state and observation models into a linear, convex framework, alleviating the need of nonlinear optimization. The framework is benchmarked against competitive variational DA baselines on two chaotic systems and a numerical weather prediction task.

**Strengths:**

- *Originality*: This work stands out as original as it avoids the DL trend and looks for an approximated solution which preserves theoretical guarantees.
- *Quality & Clarity*: Thorough background as well as theoretical analysis of the framework is provided; proving the existence of linear dynamics with consistent convergence between the original and feature space solutions.
- *Significance*: The work addresses a core task, forecasting of dynamical systems, and provides improvement in efficiency and accuracy over evaluated settings.

**Weaknesses:**

- *Baseline choices*: The authors describe many DL approaches suggested for the task and it is unclear why the performance of the chaotic system is evaluated only with respect to Frerix et al., 2021, neglecting the more recent attempts (e.g. Bocquet et al. 2024) and avoiding the Latent-x-Var baseline considered in the NWP task.
- *Efficiency claims*: The framework is compared to variational approaches which rely on GPU acceleration via JAX (Bradbury et al., 2018) however runtime was evaluated on CPU making the comparison misleading and unfair. To address this the authors can either run each framework on most suitable hardware or state this evaluation limitation in the main text. A final alternative is to provide a GPU compatible *Tensor-Var* implementation.
- *Evaluation on synthetic data*: As mentioned by the authors, showcasing performance on simulated data is a limitation of the current work. It makes it hard to evaluate the applicability and performance over real world applications.
- *Poorly written*: The paper is hard to read, it contains many grammatical errors, unclear sentences, and many repetitions. For example, lines 164-166 repeat definition mentioned a few times and contain grammatical inconsistencies, Lines 270-271 contain errors, and the entire conclusion section can be improved.
(**minor; remaining paragraphs from the template on page 10).

**Questions:**

In light of the above weaknesses, it will be valuable if the authors would:
(1) clarify baseline choices as well as runtime evaluation;
(2) showcase a more realistic application;
(3) proofread the manuscript to improve readability.

---

> ### Author Response · Authors · 2024-11-21
> **Response to Weakness**
>
> We sincerely thank the reviewer for the detailed and insightful review, as well as for recognizing the strengths of our work. Below, we provide detailed responses to your comments, identified weaknesses, and questions.
>
> ### **(1) Baseline choices**
>
> We would like to thank the reviewers for the comments and for raising concerns regarding the baseline choices. Our responses are provided below:
>
>  - In the low-dimensional cases discussed in Section 4.1, full-space 3D-Var and 4D-Var can be easily implemented without the compression error introduced by learning a latent space in Latent-x-Var. Thus, the advantages of Latent-x-Var are not apparent in these cases, and its performance is unlikely to surpass full-space x-Var in low-dimensional systems. We also compared evaluation time, highlighting one key advantage of Latent-x-Var in the high-dimensional WeatherBench setting. Additionally, we compared our method with those of Frerix et al. (2021), a popular baseline for machine learning-hybrid variational DA approaches, which still perform full-space 4D-Var by learning an inverse observation operator to fill in the missing information. Comparisons with the three chosen baselines demonstrate the effectiveness of our method in dealing with low-dimensional chaotic systems, as supported by Table 1 and Figures 8 and 9.
>  - The recent work by Bocquet et al. (2024) aims to surrogate the DA procedure by using an end-to-end supervised learning framework, with prior states and observations as inputs and the ground truth as the direct output. However, this approach lacks an online optimization mechanism, deviating from our method, which focuses on online DA with more flexible settings, such as the length of the assimilation window. Although Bocquet et al. (2024) show promise in the low-dimensional Lorenz test case, it has not yet been evaluated on high-dimensional spatiotemporal systems, making it unsuitable for direct comparison.
>  - For Latent-x-Var, the latent space is constructed using deterministic autoencoders, as done in Zhuang et al. (2022) and Cheng et al. (2024), rather than VAEs. This choice aligns with our primary goal of comparing the proposed approach with latent DA methods. VAEs, while effective for 3DVar setups (J. Mack et al., 2020; Peyron et al., 2021; Melinc et al., 2024), are generally not utilized for reduced-order modeling of chaotic systems due to the added complexity of predicting the distribution in latent space. Moreover, latent 4D-Var implementations using VAEs are rare, and we would appreciate it if the reviewer could provide related references. We implemented latent 3D-Var and 4D-Var using TorchDA, an optimized state-of-the-art PyTorch-based DA tool (Cheng et al., 2024).
>  - Additionally, we have added new experiments (see Section 4.3) incorporating more realistic observations from weather satellite tracks and much higher spatial resolution (up to 240*121) in Variational DA for Global NWP. This includes comparisons with the Latent 3D-Var and Latent 4D-Var baselines. The results demonstrate that our method consistently outperforms baseline approaches, highlighting its robustness under realistic observational conditions and showing notable improvements in assimilation accuracy for large-scale systems.
>
> We appreciate the reviewer's valuable comments and suggestions, which have helped us improve the quality of our work. We hope that our clarifications and additional experiments address your concerns.
>
>  - References:
>    - Frerix, Thomas, et al. "Variational data assimilation with a learned inverse observation operator." International Conference on Machine Learning. PMLR, 2021.
>    - Bocquet, Marc. "Surrogate modeling for the climate sciences dynamics with machine learning and data assimilation." Frontiers in Applied Mathematics and Statistics 9 (2023): 1133226.
>    - Zhuang, Yilin, et al. "Ensemble latent assimilation with deep learning surrogate model: application to drop interaction in a microfluidics device." Lab on a Chip 22.17 (2022): 3187-3202.
>    - Cheng, Sibo, et al. "Efficient deep data assimilation with sparse observations and time-varying sensors." Journal of Computational Physics 496 (2024): 112581.
>    - Mack, Julian, et al. "Attention-based convolutional autoencoders for 3d-variational data assimilation." Computer Methods in Applied Mechanics and Engineering 372 (2020): 113291.
>    - Peyron, Mathis, et al. "Latent space data assimilation by using deep learning." Quarterly Journal of the Royal Meteorological Society 147.740 (2021): 3759-3777.
>    - Melinc, Boštjan, and Žiga Zaplotnik. "3D‐Var data assimilation using a variational autoencoder." Quarterly Journal of the Royal Meteorological Society 150.761 (2024): 2273-2295.
>    - Cheng, Sibo, et al. "TorchDA: A Python package for performing data assimilation with deep learning forward and transformation functions." Computer Physics Communications 306 (2025): 109359.

---

> ### Author Response · Authors · 2024-11-21
> **Response to Weakness (Continue)**
>
> ### **(2) Efficiency claims:**
>
> We thank the reviewer for raising this concern regarding runtime evaluation. We have clearly stated the evaluation settings in the main text of the updated manuscript in Section 4.
>
> To clarify, we chose the CPU for runtime comparisons for three main reasons: (1) The baseline methods are implemented in JAX, while Tensor-Var is implemented in PyTorch. Evaluating on CPU ensures fairness by avoiding discrepancies arising from platform-specific GPU optimizations. We also optimized the baseline performance with just-in-time (JIT) compilation which significantly improves the CPU performance of Python code (Part 2, Chapter 5, Grigory Sapunov, 2024). (2) For low-dimensional problems (e.g., Lorenz and KS systems), the performance gains from GPU acceleration can be minimal because the primary computational overhead comes from the L-BFGS optimization, which cannot be parallelized effectively (Boyd et al., 2004). For the NWP task, we report evaluation times on a GPU device since all implementations use the same platform. GPU acceleration improves efficiency in handling the high-dimensional 2D image data associated with this task, and (3) JAX is a generic framework optimized for both CPU and GPU implementations, making CPU evaluations fair and appropriate for the baseline methods (Bradbury et al. 2018).
>
>   We thank the reviewer for their feedback, which helped clarify our evaluation settings. These points have been clearly stated and justified in the revised manuscript to avoid any confusion.
>
> - References
>     - Sapunov, Grigory. Deep Learning with JAX. Simon and Schuster, 2024.
>     - Bradbury, James, et al. "JAX: composable transformations of Python+ NumPy programs." (2018).
>     - Boyd, Stephen, and Lieven Vandenberghe. Convex optimization. Cambridge University Press, 2004.
>
> ### **(3) Evaluation on synthetic data**
>
> We appreciate the reviewer’s comment regarding the limitation in demonstrating applicability to real-world scenarios.
>
> In response, we have added a new experiment in Section 4.3 of the updated manuscript. Specifically, we conducted a practical global numerical weather prediction (NWP) experiment using real-time weather satellite observations at a spatial resolution of up to $240\times121$. This experiment includes comparisons with the Latent 3D-Var and Latent 4D-Var baselines. The results demonstrate that Tensor-Var achieves superior performance in terms of accuracy and robustness for large-scale systems with real-world observations, highlighting its potential for operational data assimilation and forecasting applications.
>
>  We hope this new experiment clarifies the practical relevance of our approach and addresses your concerns effectively.
>
> ### **(4) Poorly written**
>
> We thank the reviewer for carefully reading our manuscript and providing valuable feedback on its clarity and presentation. We have carefully revised the manuscript to address these issues and thoroughly proofread it to enhance clarity and consistency (all the corrections have been marked in red). We believe the revised version is now easier to follow and effectively addresses the reviewer’s concerns.

---

> ### Author Response · Authors · 2024-11-21
> **Response to Question**
>
> Thanks for your questions and suggestions. Please see our responses in weaknesses (1), (3), and (4)
>
> We appreciate the reviewer's time and effort in reviewing our manuscript. Your constructive comments and insightful suggestions have greatly helped us improve the work. If you have any further questions, we will gladly address them.

---

> > ### Comment · Reviewer_9XsZ · 2024-11-24
> > **Rebuttal period response**
> >
> > Thank you for the detailed response and significant effort put into the MS revision, I have updated my score in accordance.

---

> > > ### Author Response · Authors · 2024-11-24
> > >
> > > We would like to thank reviewer 9XsZ for thoroughly reviewing our response and recognizing our effort invested in the revision. We greatly appreciate your updated score and constructive feedback, which have been very helpful in improving our work.

---

### Official Review · Reviewer_DyoK · 2024-11-03

**Soundness:** 3
**Presentation:** 3
**Contribution:** 3
**Rating:** 6
**Confidence:** 3

**Summary:**

The paper introduces Tensor-Var, a novel data assimilation (DA) framework that uses Conditional Mean Embeddings (CME) in conjunction with deep learning based nonlinear feature maps ("deep features", DFs) to linearize observed nonlinear dynamics in feature space, addressing issues of the traditional 4D-Var data assimilation problem. Tensor-Var achieves both competitive accuracy and efficiency compared to 4D-Var and other baselines in chaotic dynamics and numerical weather prediction benchmarks.

**Strengths:**

- The authors use known and established methods in the field of deep learning and nonlinear dynamics (CME, DFs, delay embeddings, overcomplete autoencoders) to substantially improve over existing DA methods.
- The paper is generally well-written
- The authors clearly motivate each and every modification to the final employed algorithm they propose and provide plenty of theoretical justifications

**Weaknesses:**

- A lot of compartments of the approach have deep roots in dynamical systems theory (DST), which the authors do not seem to touch upon:
1. using a history of observations to better model the state of the underlying dynamical system is connected to the method of delay embeddings, often used for attractor/state space reconstruction in nonlinear dynamics [1, 2]. This field has ample literature, including (optimal) heuristics to create these embeddings, from choosing time lag between history time stamps to number of lags/dimension of the embedding [3]. This literature might provide better ways of finding an optimal representation of the underlying system state and might a reasonable alternative to costly cross-validation to find the optimal history length. All this could be discussed when introducing the history method in section 3.1.
2. Mapping nonlinear dynamics in (infinite) higher dimensional spaces with the aim to linearize the dynamics in this space is the core idea of Koopman Operator theory [4]. However, I am not familiar with KKL-observer theory (which the authors address in the paper). These fields might be strongly connected. Anyhow, at least a brief connection to this large field of research is missing in the manuscript.

- As the authors point out in their limitations: The feasibility of the framework in real-world applications is questionable due to the ground-truth state data demand. However, a more elaborate discussion on how Tensor-Var yields any improvement in this data setting compared to traditional DA methods would be welcome to guide future research.

Minor Details:
- ll. 173-174: “Equation equation 1”
- ll. 356-357: “global optimal” should be “global optimum”
- Author Contributions and Acknowledgments on p. 10 seem to contain the sample text of the ICLR template.

References:

[1] Sauer, Tim, James A. Yorke, and Martin Casdagli. "Embedology." Journal of statistical Physics 65 (1991): 579-616.

[2] Krämer, Kai-Hauke, et al. "A unified and automated approach to attractor reconstruction." New Journal of Physics 23.3 (2021): 033017.

[3] Kantz, Holger, and Thomas Schreiber. Nonlinear time series analysis. Cambridge university press, 2003.

[4] Brunton, Steven L., et al. "Modern Koopman theory for dynamical systems." arXiv preprint arXiv:2102.12086 (2021).

**Questions:**

- Is the entire loss objective of Tensor-Var really convex if the DFs are jointly learned with the linear operators/CMEs? Since the feature maps are parameterized by nonlinear NNs, the optimization should still be non-convex, no?
- Is there an intuitive reason why the NRMSE distributions of Latent 4D-Var are a lot more spread compared to Latent 3D-Var and Tensor-Var?
- Can the authors comment on the optimization of loss (7)? Learning the inverse feature map $\phi^\dagger_{\theta'_s}$ in this form comes down to an overcomplete autoencoder, which, when trained with a simple reconstruction loss, can often fall short in learning useful features and may easily overfit. Did the authors face any problems when training this architecture? And how does the regularization weight w influence the performance of Tensor-Var?
- A linear approximation of nonlinear dynamics can only be approximate, and hence, especially in chaotic systems, longer roll-outs will lead to qualitative and quantitative differences in the learned linear dynamics compared to the true dynamics (i.e. a linear model of nonlinear dynamics can not reproduce the “climate” or long-term behavior of the underlying system). Can the authors comment on this in the context of their DA approach?

I'm happy to increase my score if the authors appropriately address my comments and questions.

---

> ### Author Response · Authors · 2024-11-21
> **Response to Weakness**
>
> We sincerely thank you for your detailed and insightful review, as well as for highlighting the strengths of our work. Below, we address your comments, weaknesses, and questions in detail:
>
> ### **(1) Discussion and reviews on history embedding.**
> - We thank the reviewer for this insightful suggestion on the literature on delay embeddings and have provided a summary of the discussion in the updated manuscript, see Lines 210-215.
>
>    As the reviewer pointed out, we use delay embeddings to incorporate a history of observations, improving the modeling of the underlying dynamical system's state from partial observations. Delay embedding is a well-established method in nonlinear dynamics for reconstructing attractors and state spaces from sequential data (Sauer et al., 1991; Krämer et al., 2021). Based on Whitney's embedding theorem and Takens' seminal work (Floris Takens, 2006), Takens' theorem demonstrates that the dynamics of a system can be reconstructed in a higher-dimensional space using delay coordinates.
>
>    Recent theoretical advancements parallel to Takens' embedding theorem have focused on partial observability from a learning theory perspective (Uehara et al., 2022). showed that the required history length for effective state reconstruction is determined by the complexity of the dynamical system such as system dimensions. These results provide a guide to determine embedding parameters, i.e., history length, instead of intensive cross-validation. We provide an ablation study on the robustness of these length choices in Section 4.4.
>
>    We incorporate this principle using tensor-product kernel features (Song et al., 2009) between observations and history, allowing us to better capture high-order complex dependencies in the DA system.
>
> - References
>   - Song, Le, et al. "Hilbert space embeddings of conditional distributions with applications to dynamical systems." Proceedings of the 26th Annual International Conference on Machine Learning. 2009.
>   - Sauer, Tim, James A. Yorke, and Martin Casdagli. "Embedology." Journal of Statistical Physics 65 (1991): 579-616.
>   - Krämer, Kai-Hauke, et al. "A unified and automated approach to attractor reconstruction." New Journal of Physics 23.3 (2021): 033017.
>   - Takens, Floris. "Detecting strange attractors in turbulence." Dynamical Systems and Turbulence, Warwick 1980: proceedings of a symposium held at the University of Warwick 1979/80. Berlin, Heidelberg: Springer Berlin Heidelberg, 2006.
>   - Uehara, Masatoshi, et al. "Provably efficient reinforcement learning in partially observable dynamical systems." Advances in Neural Information Processing Systems 35 (2022): 578-592.
>   - Liu, Qinghua, et al. "When is partially observable reinforcement learning not scary?." Conference on Learning Theory. PMLR, 2022.
>
> ### **(2) Discussions on Koopman operator theory and KKL observer theory**
> - We thank the reviewer for this valuable comment on the connection between KKL observer theory and Koopman operator theory. Our response is below:
>
>   As the reviewer reconginzied, the KKL observer theory indeed shares a natural and deep connection with the Koopman operator theory, both aiming to use linear structures to analyze nonlinear systems.
>
>   The Koopman operator (Koopman, 1931; Bevanda et al., 2021) provides a linear view of nonlinear system dynamics by representing the evolution of observables in an infinite-dimensional function space, rather than acting directly on state variables. Similarly, the KKL observer (Andrieu et al., 2006) linearizes nonlinear state dynamics into a higher-dimensional feature space, transforming the system into a linear PDE to enable effective state estimation through linear observer design. In this feature space, state estimation errors are consistent with those in the original space. Our method follows the principle in KKL observer theory. Our method transforms state dynamics into a linear PDE using a kernel feature map (see our analysis in Appendix A). This enables consistent state estimation and leverages the provided linear properties to improve the accuracy and efficiency of analyzing nonlinear dynamics.
>
> - References
>   - Koopman, Bernard O. "Hamiltonian systems and transformation in Hilbert space." Proceedings of the National Academy of Sciences 17.5 (1931): 315-318.
>   - Bevanda, Petar, Stefan Sosnowski, and Sandra Hirche. "Koopman operator dynamical models: Learning, analysis and control." Annual Reviews in Control 52 (2021): 197-212.
>   - Andrieu, Vincent, and Laurent Praly. "On the existence of a Kazantzis--Kravaris/Luenberger observer." SIAM Journal on Control and Optimization 45.2 (2006): 432-456.

---

> ### Author Response · Authors · 2024-11-21
> **Response to Weakness (Continue)**
>
> ### **(3) Limitations in real-world applications**
>
> - We thank the reviewer for this precious suggestion and agree that a discussion on the improvements of Tensor-Var in the synthetic setting would help enhance the clarity and impact of our work.
>
>   - **Experiment limitations.** In our added experiments,  we consider a more realistic DA problem in global NWP by assimilating observations from satellite tracks with a higher spatial resolution ($240\times121$). This scenario poses significant challenges, including the dynamic nature of observation locations and the spatial-temporal sparsity of data along satellite tracks. Our results show that Tensor-Var consistently outperforms the two Latent-DA baselines. The results demonstrate that Tensor-Var achieves superior performance in terms of accuracy and robustness for large-scale systems with real-world observations, highlighting its potential for operational data assimilation and forecasting applications. The details of the settings and results can be found in Section 4.3 of the updated manuscript, and we will elaborate further in the camera-ready version.
>
>   - **Comparison to traditional DA.** Our work is motivated by the challenging nonlinear optimization inherent in the 4D-Var method and draws inspiration from advancements in representation learning using modern kernel methods. Key challenges include (1) the high nonlinearity (and computational cost) of dynamical and observational models and (2) the high dimensionality of the DA system.
>
>     Compared to traditional Var-DA approaches, our method adopts a data-driven paradigm, offering greater computational efficiency by avoiding the reliance on numerical models. The key distinction from existing data-driven Var-DA methods lies in our "linearisation" strategy, which simplifies the optimisation process from two perspectives: (1) We solve the 4D-Var optimisation problem entirely within a low-dimensional feature space, avoiding projection back to the original space and the large-scale automatic differentiation required by Xu et al., (2024) and Cheng et al., (2024), thus reducing computational overhead. (2) The linear structure of the low-dimensional feature space admits a convex cost function, offering faster convergence and numerical stability within the feature space, as demonstrated in our experiments.
>
>   Building on existing work, we also note that the kernel methods have been extensively explored in Kalman-filtering-based DA approaches, where they have shown promise in capturing nonlinearities with linear kernel operators, see Song et al., (2009) and Mauran et al., (2023). However, to the best of our knowledge, our work is the first to integrate kernel methods into a variational DA framework. Leveraging recent advancements in deep kernel representations (Xu et al., 2022; Shimizu et al., 2024), our approach reduces the cubic computational cost typically associated with kernel methods. See more details in section 3.2. This integration bridges the gap between kernel-based representation learning and Var-DA, paving a new way for exploring the applications of modern kernel methods and advancing efficient solutions for high-dimensional DA challenges.
>
> - References
>    - Xu, Xiaoze, et al. "Fuxi-DA: A Generalized Deep Learning Data Assimilation Framework for Assimilating Satellite Observations." arXiv preprint arXiv:2404.08522 (2024).
>    - Cheng, Sibo, et al. "TorchDA: A Python package for performing data assimilation with deep learning forward and transformation functions." Computer Physics Communications 306 (2025): 109359.
>    - Shimizu, Eiki, Kenji Fukumizu, and Dino Sejdinovic. "Neural-Kernel Conditional Mean Embeddings." arXiv preprint arXiv:2403.10859 (2024).
>    - Xu, Liyuan, et al. "Importance weighted kernel Bayes’ rule." International Conference on Machine Learning. PMLR, 2022.
>    - Song, Le, et al. "Hilbert space embeddings of conditional distributions with applications to dynamical systems." Proceedings of the 26th Annual International Conference on Machine Learning. 2009.
>    - Mauran, Sophie, et al. "A kernel extension of the Ensemble Transform Kalman Filter." International Conference on Computational Science. Cham: Springer Nature Switzerland, 2023.
>
> ### **(4) Minor details**
>
> - We thank the reviewer for carefully reading our manuscript. All the identified typos have been corrected, and we have done proofreading to ensure clarity and consistency throughout the manuscript.

---

> ### Author Response · Authors · 2024-11-22
> **Response to Question**
>
> ### **(1) Is the entire loss objective of Tensor-Var really convex if the DFs are jointly learned with the linear operators/CMEs? Since the feature maps are parameterized by nonlinear NNs, the optimization should still be non-convex, no?**
>
> - We thank the reviewer for raising this important question regarding the convexity of the cost function in Tensor-Var.
>
>
>  -  To clarify, the convexity of the cost function in Tensor-Var applies only within the low-dimensional feature space once the feature maps (DFs) are fixed after training. As correctly pointed out by the reviewer, the overall 4D-Var cost function remains non-convex due to the nonlinear parameterization introduced by the neural networks. As detailed in section 3.1, our linearization strategy admits a linear DA system in the feature space and the cost function becomes convex within the feature space. This results in faster convergence and improved numerical stability during this stage of optimization. This distinction is crucial and highlights a key advantage of our approach compared to fully nonlinear alternatives.
>
>  -  We have revised our manuscript to clarify the convexity and further elaborate on the implications of our linearisation strategy for overall performances. Thank you again for pointing out this important question to improve the clarity of our work.
>
> ### **(2) Is there an intuitive reason why the NRMSE distributions of Latent 4D-Var are a lot more spread compared to Latent 3D-Var and Tensor-Var?**
>
> - The broader spread in the NRMSE distributions of Latent 4D-Var can be attributed to the reason: Unlike static Latent 3D-Var, Latent 4D-Var optimizes over a temporally extended window, which introduces extra sensitivity to errors in the fitted dynamical model across multiple time steps. Specifically, the latent 4D-Var build the dynamical model on a latent space derived from the auto-encoder used in 3D-Var. As the reviewer pointed out in next question, the standard auto-encoders with reconstructuion often overfit, leading to a poorly-structured latent space that makes fitting an accurate dynamical model for complex system challenging. We will provide further explanations and describe how Tensor-Var addresses this issue in the next answer.
>
> ### **(3) Can the authors comment on the optimization of loss (7)? Learning the inverse feature map in this form comes down to an overcomplete autoencoder, which, when trained with a simple reconstruction loss, can often fall short in learning useful features and may easily overfit. Did the authors face any problems when training this architecture? And how does the regularisation weight w influence the performance of Tensor-Var?**
>
> We thank the reviewer for this insightful question regarding the optimisation of loss (7) and the potential challenges of overfitting in autoencoders. We did not encounter significant issues when training the DFs but agree the reviewer that the penalty coefficient $w$ can positively or negatively influence performance depending on its tuning.
>
> Unlike Latent-DA, which relies solely on a reconstruction loss, Tensor-Var jointly learns feature maps (DFs) and identify the dynamical system through the two terms in loss (7). Importantly, unlike existing data-driven system identification methods, we do not explicitly parameterize the dynamical model as a fully-connected (FC) layer. Instead, we use the analytical form of the kernel CME operator as stated in Equation (3). This approach offers several advantages: (1) By incorporating the dynamics into the learning process, Tensor-Var captures a feature space better aligned with the underlying system (Miyato et al., 2022; Koyama et al., 2023). (2) Since the dynamical model is not over-parameterised with FC layer (i.e. a dense matrix), the risk of overfitting is reduced (Salman et al., 2019, Srivastava et al., 2014).
>
> - References
>     - Salman, Shaeke, and Xiuwen Liu. "Overfitting mechanism and avoidance in deep neural networks." arXiv preprint arXiv:1901.06566 (2019).
>     - Srivastava, Nitish, et al. "Dropout: a simple way to prevent neural networks from overfitting." The journal of machine learning research 15.1 (2014): 1929-1958.
>     - Miyato, Takeru, Masanori Koyama, and Kenji Fukumizu. "Unsupervised learning of equivariant structure from sequences." Advances in Neural Information Processing Systems 35 (2022): 768-781.
>     - Koyama, Masanori, et al. "Neural fourier transform: A general approach to equivariant representation learning." arXiv preprint arXiv:2305.18484 (2023).

---

> > ### Author Response · Authors · 2024-11-22
> > **Response to Question (Continue)**
> >
> > ### **(4) A linear approximation of nonlinear dynamics can only be approximate, and hence, especially in chaotic systems, longer roll-outs will lead to qualitative and quantitative differences in the learned linear dynamics compared to the true dynamics (i.e. a linear model of nonlinear dynamics can not reproduce the “climate” or long-term behavior of the underlying system). Can the authors comment on this in the context of their DA approach?**
> >
> > - We thank the reviewer for raising the question regarding the linear approximation.
> > - The use of linear representations for modeling chaotic systems is well-supported by numerous studies (Neumann, et al., 1932, Hou, et al., 2023; Colbrook, et al., 2024;), with preserving long-term statistics(behaviour) effectively. Linear representation-based approaches have demonstrated strong performance in various data-driven scenarios ( Budišić, et al., 2012; Giannakis, et al., 2018; Rice, et al., 2020) and can theoretically serve as universal approximators of dynamical systems (Tian, et al., 2021; Jia Guo, 2021; Kostic, et al., 2021; Bevanda, et al., 2024). This is closely related to Koopman operator theory and KKL observer theory, as the reviewer pointed out and we discussed in our response to weakness (2).
> > - Importantly, the linear representation refers to be in function space, projected by combination of nonlinear functions (the feature map in our approach). Therefore, our model is not purely linear—the linearity only applies to the dynamics within the feature space, where feature maps are defined by nonlinear kernels or neural networks. These nonlinear feature maps capture capture highly nonlinear patterns of the dynamical system, aligning with the well-founded theoretical frameworks.
> >
> > - In the context of DA, long-term forecasting ability is not a priority for the following reasons:
> >    - The assimilation window is usually short, meaning that precise long-term forecasting is not required from our model.
> >    - As assimilation progresses, new observations become available, and Tensor-Var performs data assimilation to update the state estimates continuously, providing the accurate short/mid-term forecast in a cyclical manner.
> >
> > - References
> >    - Tian, Wenchong, and Hao Wu. "Kernel embedding based variational approach for low-dimensional approximation of dynamical systems." Computational Methods in Applied Mathematics 21.3 (2021): 635-659.
> >    - Guo, Jia. "Convergence of kernel methods for modeling and estimation of dynamical systems." (2021).
> >    - Bevanda, Petar, et al. "Koopman kernel regression." Advances in Neural Information Processing Systems 36 (2024).
> >    - Kostic, Vladimir, et al. "Learning dynamical systems via Koopman operator regression in reproducing kernel Hilbert spaces." Advances in Neural Information Processing Systems 35 (2022): 4017-4031.
> >    - Colbrook, Matthew J., Catherine Drysdale, and Andrew Horning. "Rigged Dynamic Mode Decomposition: Data-Driven Generalized Eigenfunction Decompositions for Koopman Operators." arXiv preprint arXiv:2405 00782 (2024).
> >     - Hou, Boya, et al. "Sparse learning of dynamical systems in RKHS: An operator-theoretic approach." International Conference on Machine Learning. PMLR, 2023.
> >     - Neumann, J. V. "Proof of the quasi-ergodic hypothesis." Proceedings of the National Academy of Sciences 18.1 (1932): 70-82.
> >     - Rice, Julian, Wenwei Xu, and Andrew August. "Analyzing Koopman approaches to physics-informed machine learning for long-term sea-surface temperature forecasting." arXiv preprint arXiv:2010.00399 (2020).
> >     - Giannakis, Dimitrios, et al. "Koopman analysis of the long-term evolution in a turbulent convection cell." Journal of Fluid Mechanics 847 (2018): 735-767.
> >     - Budišić, Marko, Ryan Mohr, and Igor Mezić. "Applied koopmanism." Chaos: An Interdisciplinary Journal of Nonlinear Science 22.4 (2012).
> >
> >
> > We thank the reviewer again for the insightful comments and constructive feedback, which greatly help us improve the quality of our work. We hope these clarifications address the concerns and questions raised. We look forward to your further comments and feedback.

---

> ### Author Response · Authors · 2024-11-24
>
> May I ask the reviewer whether our response/improvement made has addressed your concerns? There is not much time left for discussion. We would be pleased to answer your further questions. Many thanks.

---

> > ### Comment · Reviewer_DyoK · 2024-11-24
> >
> > I thank the authors for their detailed rebuttal, especially for including a real-world example. Since my concerns and questions have been addressed, I have updated my score.

---

> > > ### Author Response · Authors · 2024-11-24
> > >
> > > We would like to thank reviewer DyoK for recognising our efforts and updated score. We appreciate your re-evaluation and are glad that our response addressed your concerns.

---

### Official Review · Reviewer_JYJC · 2024-11-04

**Soundness:** 2
**Presentation:** 3
**Contribution:** 3
**Rating:** 8
**Confidence:** 3

**Summary:**

This paper proposes an innovative Tensor-Var method, which integrates kernel conditional mean embedding with 4D-Var data assimilation to address computational efficiency and convergence issues in the optimization of nonlinear dynamical systems.

**Strengths:**

The Tensor-Var method is proposed, applying kernel conditional mean embedding to 4D-Var data assimilation, significantly improving the optimization efficiency of nonlinear dynamical systems.

**Weaknesses:**

Although the method shows remarkable performance on simulated data, its validation on real-world datasets is relatively limited. Additional tests in practical application scenarios could be supplemented.

**Questions:**

see Weaknesses

---

> ### Author Response · Authors · 2024-11-21
> **Response to Weakness**
>
> We thank the reviewer for spending time and effort in reviewing our paper. We appreciate you recognising our work's strengths and the constructive comment on the limited validation in real-world DA problems.
>
> In response, we added a practical global numerical weather prediction (NWP) experiment using real-time weather satellite observations at a spatial resolution of up to 240*121, including comparisons with the Latent 3D-Var and Latent 4D-Var baselines. The results demonstrated that Tensor-Var achieved superior performance in terms of accuracy and robustness for large-scale systems with real-world observations, highlighting its potential for operational data assimilation and forecasting applications. The detailed experimental settings and results have been included in the revised manuscript (see Section 4.3).
>
> We hope that the added experiment and corresponding results sufficiently address your concerns, and we look forward to your further feedback.

---

> ### Author Response · Authors · 2024-11-24
>
> May I ask the reviewer whether our response/improvement made has addressed your concerns? There is not much time left for discussion. We would be pleased to answer your further questions. Many thanks.

---

> ### Author Response · Authors · 2024-11-25
>
> This is a gentle reminder to inquire for reviewer JYJC if our additional real-world application results have addressed your concerns. Please note that the deadline is approaching, and we would greatly appreciate it if you could share any further feedback or confirm acceptance of our rebuttal.
>
> Thank you for your time and consideration.

---

> > ### Comment · Reviewer_JYJC · 2024-11-25
> >
> > I am so happy to see the experiments added by the author, which adequately address my query, and I would be willing to raise my rating and recommend acceptance of the change of paper

---

> > > ### Author Response · Authors · 2024-11-25
> > >
> > > We would like to thank reviewer JYJC for the constructive feedback and acknowledging the added experiments. We are glad to hear that our responses with the new experiment adequately address your query. Your re-evaluation and decision to raise the score are greatly appreciated.

---

### Author Response · Authors · 2024-11-21
**General Response**

We sincerely thank all reviewers for your time and effort in reviewing our paper. We appreciate your recognition of the novelty and strengths of our work. Your insightful comments have greatly improved our manuscript, and we will acknowledge your contributions to the camera-ready version.

### To clarify, we have restated our contributions as follows:

- **Novel Framework:**  We introduced the kernel conditional mean embedding framework to variational data assimilation for the first time, offering a principled approach to linearize DA systems with a convex 4D-Var cost function in the feature space.
- **Handling Incomplete Observations:** To address challenges posed by incomplete observations, we derived an inverse observation operator that leverages historical data to infer the system state, using a tensor-product approach that captures high-order correlations.
- **Theoretical Consistency:** We provided theoretical proof that the 4D-Var optimization solution in the feature space is consistent with optimisation in the original space.
- **Scalability and Comprehensive Experiments:** We enhanced the scalability of traditional CME methods by adaptively learning deep kernel features. This was validated through comparisons with popular baselines—including operational Variational DA and ML-hybrid Variational DA—across a comprehensive set of experiments involving real-world observations at a spatial resolution of up to $240\times121$.

### We have revised our manuscript mainly in two areas (all major changes are highlighted in red in the updated manuscript):

- **Conciseness and Simplification of Section 3**: We reduced the number of complex mathematical equations and notations to improve readability.
- **Addition of a Global NWP Experiment with practical observations in Section 4.3**: We added a practical global numerical weather prediction (NWP) experiment using real-time weather satellite observations at a spatial resolution of up to 240*121. This includes comparisons with the Latent 3D-Var and Latent 4D-Var baselines. The results demonstrate that Tensor-Var achieves robust performance in terms of accuracy and robustness for large-scale systems with real-world observations, highlighting its potential for operational data assimilation and forecasting applications.

Once again, we thank all the reviewers for their valuable suggestions and feedback. We look forward to your further comments and will promptly address any additional questions you may have.

---

### Note · Authors · 2025-01-31

I have read and agree with the venue's withdrawal policy on behalf of myself and my co-authors.